# Comparative genomics reveals insight into the evolutionary origin of massively scrambled genomes

Yi Feng[1], Rafik Neme[1,2], Leslie Y Beh[1], Xiao Chen[3], Jasper Braun[4†], Michael W Lu[1], Laura F Landweber[1]*

[1]Departments of Biochemistry and Molecular Biophysics and Biological Sciences, Columbia University, New York, United States; [2]Department of Chemistry and Biology, Universidad del Norte, Barranquilla, Colombia; [3]Pacific Biosciences, Menlo Park, United States; [4]Department of Mathematics and Statistics, University of South Florida, Tampa, United States

*For correspondence: Laura.Landweber@columbia.edu

Present address: †Department of Pathology, Beth Israel Deaconess Medical Center, Boston, United States

**Abstract** Ciliates are microbial eukaryotes that undergo extensive programmed genome rearrangement, a natural genome editing process that converts long germline chromosomes into smaller gene-rich somatic chromosomes. Three well-studied ciliates include *Oxytricha trifallax*, *Tetrahymena thermophila,* and *Paramecium tetraurelia*, but only the *Oxytricha* lineage has a massively scrambled genome, whose assembly during development requires hundreds of thousands of precisely programmed DNA joining events, representing the most complex genome dynamics of any known organism. Here we study the emergence of such complex genomes by examining the origin and evolution of discontinuous and scrambled genes in the *Oxytricha* lineage. This study compares six genomes from three species, the germline and somatic genomes for *Euplotes woodruffi*, *Tetmemena sp.*, and the model ciliate *O. trifallax*. We sequenced, assembled, and annotated the germline and somatic genomes of *E. woodruffi*, which provides an outgroup, and the germline genome of *Tetmemena sp*. We find that the germline genome of *Tetmemena* is as massively scrambled and interrupted as *Oxytricha*'s: 13.6% of its gene loci require programmed translocations and/or inversions, with some genes requiring hundreds of precise gene editing events during development. This study revealed that the earlier diverged spirotrich, *E. woodruffi*, also has a scrambled genome, but only roughly half as many loci (7.3%) are scrambled. Furthermore, its scrambled genes are less complex, together supporting the position of *Euplotes* as a possible evolutionary intermediate in this lineage, in the process of accumulating complex evolutionary genome rearrangements, all of which require extensive repair to assemble functional coding regions. Comparative analysis also reveals that scrambled loci are often associated with local duplications, supporting a gradual model for the origin of complex, scrambled genomes via many small events of DNA duplication and decay.

## Editor's evaluation

The study marks a significant advance in the field of evolutionary genomics of ciliates, an ancient and highly diverse eukaryotic phylum with many idiosyncrasies that teach us valuable lessons, inter alia, about sex and the plasticity of genomes. By focusing on two species from the same family, plus a more distant outgroup within the same class, this valuable study provides new and compelling information on evolutionary trends of genome rearrangement among different species of this interesting group of organisms. The work will be of interest to anyone interested in genome dynamics.

## Introduction

Organisms do not always contain a single, static genome. Programmed genome editing is a naturally occurring and essential part of development in many organisms, including ciliates (*Chen et al., 2014*), nematodes (*Mitreva et al., 2005*), lampreys (*Smith et al., 2012*), and zebra finches (*Biederman et al., 2018*). Most of these events involve precise removal and rejoining of large regions of DNA during postzygotic differentiation of a somatic genome from a germline genome. Ciliates are microbial eukaryotes with two types of nuclei: a somatic macronucleus (MAC) that differentiates from a germline micronucleus (MIC). In the model ciliate *Oxytricha*, the MAC is entirely active chromatin (*Beh et al., 2019*) and the hub of transcription. The three species that we compare are all spirotrichs, which have gene-sized 'nanochromosomes' in the MAC, present at high copy number (*Swart et al., 2013*; *Chen et al., 2015*; *Wang et al., 2016*; *Chen et al., 2019*; *Lindblad et al., 2019*; *Vinogradov et al., 2012*). The diploid MIC participates in sexual reproduction, but its megabase-sized chromosomes are mostly transcriptionally silent.

Gene loci are often arranged discontinuously in the MIC, with short genic segments called macronuclear destined sequences (MDSs), interrupted by stretches of non-coding DNA called internally eliminated sequences (IESs) (*Figure 1A*). During sexual development, a new MAC genome rearranges from a copy of the zygotic MIC genome. MDSs join in the correct order and orientation, whereas MIC-limited genomic regions undergo programmed deletion, including repetitive elements, intergenic regions, and IESs (*Figure 1A*). Though analogous to intron splicing, these events occur on DNA. The MDSs for some MAC chromosomes are *scrambled* if they require translocation or inversion during MAC development (*Figure 1A*). Pairs of short repeats, called *pointers*, are present at MDS-IES junctions in both scrambled and nonscrambled loci (*Mitcham et al., 1992*; *Prescott, 1994*). Pointer sequences are present twice in the MIC, at the end of MDS *n* and the beginning of MDS n+1. One copy of the repeat is retained at each MDS-MDS junction in a mature MAC chromosome (*Figure 1A*). These microhomologous regions help guide MDS recombination, but most are non-unique, and the shortest pointers are just 2 bp. Thousands of long, noncoding template RNAs collectively program MDS joining (*Nowacki et al., 2008*; *Lindblad et al., 2017*; *Yerlici and Landweber, 2014*).

Numerous studies have inferred the possible scope of genome rearrangement in different ciliate species using partial genome surveys. In *Paramecium*, PiggyMac-depleted cells fail to remove MIC-limited regions properly, which provided a resource to annotate ~45,000 IESs prior to assembly of a draft MIC genome (*Arnaiz et al., 2012*). The use of single-cell sequencing has allowed pilot studies to sample partial MIC genomes of diverse species (*Chen et al., 2019*; *Maurer-Alcalá et al., 2018a*; *Maurer-Alcalá et al., 2018b*; *Smith et al., 2020*). Alignment of tentative MIC reads to either assembled MAC genomes or single-cell transcriptome data predicts over 20 candidate scrambled loci in two basal ciliates, *Loxodes sp.* and *Blepharisma americanum* (*Maurer-Alcalá et al., 2018b*) and hundreds of candidate loci in the tintinnid *Schmidingerella arcuata* (*Smith et al., 2020*). Nearly one-third (31%) of approximately 5000 surveyed transcripts may be scrambled in *Chilodonella uncinata* (*Maurer-Alcalá et al., 2018a*, *Figure 1B*), which has four confirmed cases of scrambled genes (*Katz and Kovner, 2010*; *Gao et al., 2014*). Transcriptome-based surveys offer less precise estimates and cannot distinguish RNA splicing. Several computational pipelines have been developed to facilitate the inference of genome rearrangement features by split-read mapping in the absence of complete MIC or MAC reference genomes (*Denby Wilkes et al., 2016*; *Zheng et al., 2020*; *Feng et al., 2020*; *Seah et al., 2021*). By surveying lighter genome coverage prior to full sequencing, these tools provide partial insight into germline architecture. This helps guide selection of species for full genome sequencing and subsequent construction of complete rearrangement maps between the MIC and MAC genomes. High-quality MIC genome reference assemblies are only currently available for three ciliate genera: *Oxytricha* (*Chen et al., 2014*), *Tetrahymena* (*Hamilton et al., 2016*), and *Paramecium* (*Guérin et al., 2017*; *Sellis et al., 2021*).

Programmed genome rearrangements in *Oxytricha* exhibit the highest accuracy and largest scale of any known natural gene-editing system, with exquisite control over hundreds of thousands of precise DNA cleavage/joining events. Accordingly, its germline genome structure is arguably the most complex of any model organism (*Chen et al., 2014*), requiring programmed deletion of over 90% of the germline DNA during development and massive descrambling of the resulting fragments to construct a new MAC genome of over 18,000 chromosomes (*Lindblad et al., 2019*). This differs from the distantly related *Tetrahymena* and *Paramecium* that both eliminate ~30% of the germline genome

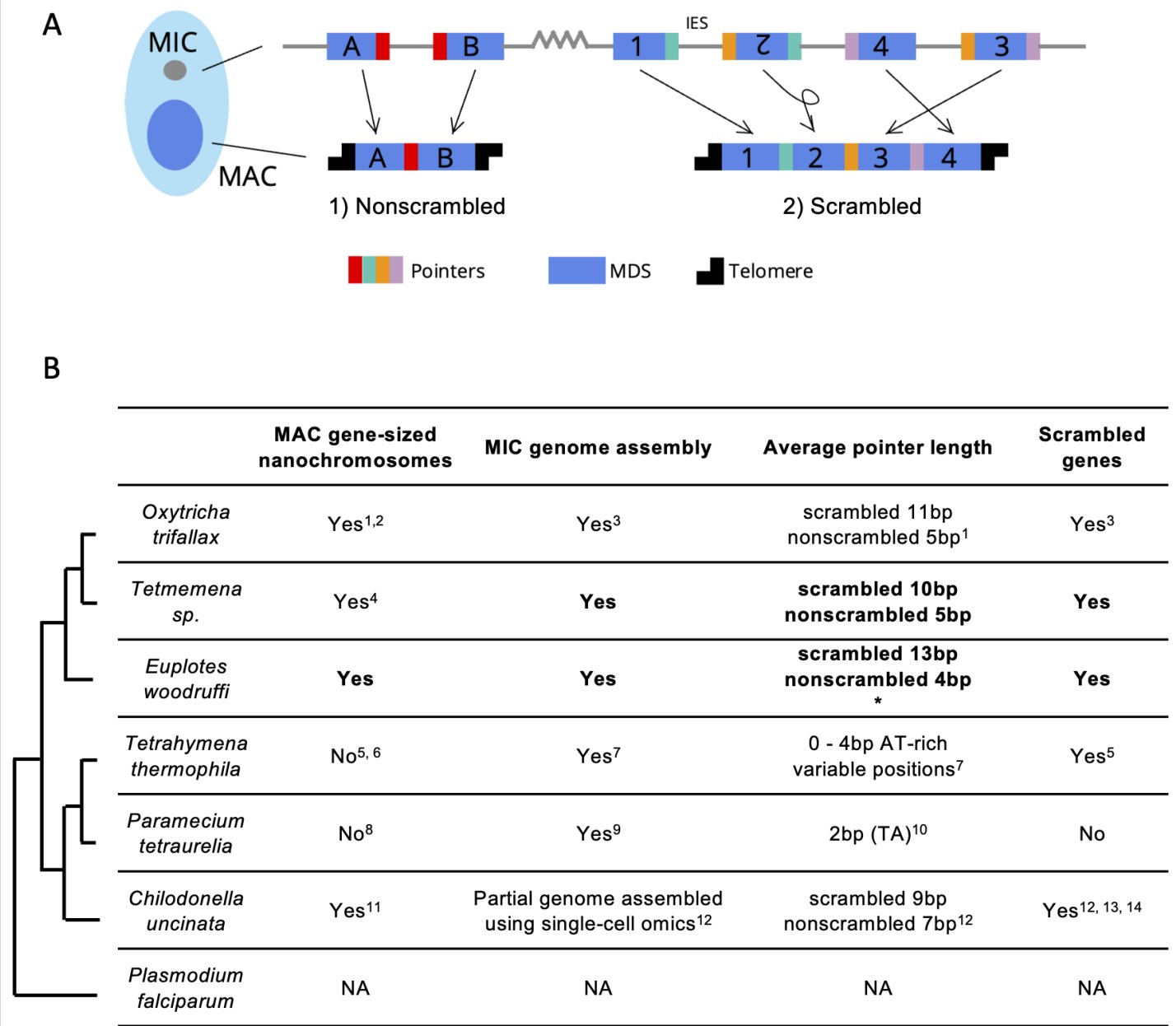

**Figure 1.** Genome rearrangements in representative ciliate species. (**A**) Diagram of genome rearrangement in *Oxytricha*. Each ciliate cell contains a somatic macronucleus (MAC) and a germline micronucleus (MIC). During development, the MAC genome rearranges from a copy of the MIC genome. (1) Nonscrambled genes rearrange simply by joining consecutive macronuclear destined sequences (MDSs, blue boxes) and removing internal eliminated sequences (IESs, thin lines). (2) Rearrangement of scrambled genes requires MDS translocation and/or inversion. Pointers are microhomologous sequences (colored vertical bars) present in two copies in the MIC and only one copy in the MAC where consecutive MDSs recombine. (**B**) Comparison of genome rearrangement features of representative ciliates and the non-ciliate *Plasmodium falciparum* as an outgroup (phylogenetic information is based on *Parfrey et al., 2011*; *Bracht et al., 2013*). Conclusions from this study are shown in bold. * indicates that some scrambled pointers in *Euplotes woodruffi* are much longer, as discussed in the results. Statistics for pointers ≤30 bp in *E. woodruffi* are shown. Table information derives from the following sources: 1 - *Swart et al., 2013*; 2 - *Lindblad et al., 2019*; 3 - *Chen et al., 2014*; 4 - *Chen et al., 2015*; 5 - *Sheng et al., 2020*; 6 - *Eisen et al., 2006*; 7 - *Hamilton et al., 2016*; 8 - *Aury et al., 2006*; 9 - *Guérin et al., 2017*; 10 - *Arnaiz et al., 2012*; 11 - *Riley and Katz, 2001*; 12 - *Maurer-Alcalá et al., 2018a*; 13 - *Katz and Kovner, 2010*; 14 - *Gao et al., 2014*.

(*Hamilton et al., 2016*; *Guérin et al., 2017*). *Paramecium* uses exclusively 2 bp pointers and lacks evidence of any scrambled loci. A small number of scrambled loci (4 confirmed out of 2711 candidates) have been reported in *Tetrahymena* (*Sheng et al., 2020*, *Figure 1B*). *Tetrahymena* and *Paramecium* diverged from *Oxytricha* over 1 billion years ago (*Parfrey et al., 2011*; *Bracht et al., 2013*),

which leaves a large gap in our understanding of the emergence of complex DNA rearrangements in the *Oxytricha* lineage.

Open questions include how did the *Oxytricha* germline genome acquire its high number of IES insertions and how do scrambled loci arise and evolve. Three previous studies tackled these questions at the level of single genes and orthologs, including DNA polymerase α, actin I, and Telomere end-binding protein subunit α (*Hogan et al., 2001*; *Chang et al., 2005*; *Wong and Landweber, 2006*; *DuBois and Prescott, 1995*). Here, we provide the first comparative genomic analysis of *Oxytricha trifallax* and two other spirotrichous ciliates, *Tetmemena sp.* and *Euplotes woodruffi. Tetmemena sp.* is a hypotrich similar to *Tetmemena pustulata*, formerly *Stylonychia pustulata* (*Chen et al., 2015*), in the same family as *O. trifallax* (*Figure 1B*; *Chen et al., 2014*; *Chen et al., 2015*). Hypotrichs are noted for the presence of scrambled genes, based on previous ortholog comparisons (*Chen et al., 2015*; *Hogan et al., 2001*; *Chang et al., 2005*; *DuBois and Prescott, 1995*; *Figure 1B*). *E. woodruffi,* together with the hypotrichous ciliates, belong to the class Spirotrichea (*Figure 1B*). Like hypotrichs, *Euplotes* also has gene-sized nanochromosomes in the MAC genome (*Wang et al., 2016*; *Chen et al., 2019*; *Chen et al., 2021*), but this outgroup uses a different genetic code (UGA is reassigned to cysteine, *Meyer et al., 1991*), and little is known about its MIC genome. A partial MIC genome of *Euplotes vannus* was previously assembled, and it contains highly conserved TA pointers (*Chen et al., 2019*), consistent with previous observations in *Euplotes crassus* (*Klobutcher and Herrick, 1995*). This differs from *O. trifallax*, which uses longer pointers of varying lengths, with scrambled pointers typically longer than nonscrambled ones (*Chen et al., 2014*, *Figure 1B*). This observation suggests that longer pointers may supply more information to facilitate MDS descrambling, sometimes over great distances. Therefore, the preponderance of 2 bp pointers in the other *Euplotes* species could indicate limited capacity to support scrambled genes, and a partial genome survey of *E. vannus* concluded that at least 97% of loci are nonscrambled (*Chen et al., 2019*). Early studies of *Euplotes octocarinatus*, on the other hand, demonstrated its use of longer pointers (that usually contain TA) (*Tan et al., 1999*; *Wang et al., 2005*), suggesting that some members of the *Euplotes* genus may have the capacity to support complex genome reorganization. To investigate the origin of scrambled genomes, we choose *E. woodruffi* as an outgroup, because it is closely related to *E. octocarinatus* (*Syberg-Olsen et al., 2016*) and feasible to culture in the lab.

This study includes the de novo assemblies of the micronuclear genome of *Tetmemena sp.* and both genomes of *E. woodruffi*. The availability of MIC and MAC genomes for both species allows us to annotate and compare their genome rearrangement maps and other key features to each other and to *O. trifallax*. The MIC genome of *Tetmemena* is extremely interrupted, like *Oxytricha*. While the *E. woodruffi* MIC genome is much more IES-sparse, it contains thousands of scrambled genes, whose architecture we compare to orthologous loci in the other species. We infer that the evolutionary origin of scrambled genes is associated with local duplications, providing strong support for a previously proposed simple evolutionary model requiring only duplication and decay (*Gao et al., 2015*) that allows for the evolutionary expansion of extremely rearranged chromosome architectures.

## Results
### Germline genome expansion via repetitive elements

*Tetmemena sp.* and *E. woodruffi* were both propagated in laboratory culture from single cells. The *E. woodruffi* MAC genome was sequenced and assembled from paired-end Illumina reads from whole cell DNA, which is mostly MAC-derived. For comparative analysis, the MAC genome of *E. woodruffi* was assembled using the same pipeline previously used for *Tetmemena sp.* (*Chen et al., 2015*). Because MIC DNA is significantly more sparse than MAC DNA in individual cells (*Prescott, 1994*), MIC DNA was enriched before sequencing (see Methods); however, this leads to much lower sequence coverage of the MIC than the MAC. Third-generation long reads (Pacific Biosciences and Oxford Nanopore Technologies) were combined with Illumina paired-end reads (Methods, see genome coverage in *Supplementary file 1*) to construct hybrid genome assemblies for *Tetmemena sp.* and *E. woodruffi*. Though the final genome assemblies are still fragmented, often due to transposon or other repetitive insertions at boundaries (*Figure 2—figure supplement 1*), the current draft assemblies cover most (>90%) MDSs for 89.1% of MAC nanochromosomes in *Tetmemena*, and for 90.0% of MAC nanochromosomes in *E. woodruffi*. This allowed us to establish near-complete rearrangement

**Table 1.** Statistics of somatic macronucleus (MAC) and germline micronucleus (MIC) genomes in three species.

| | *Oxytricha trifallax* | | *Tetmemena sp.* | | *Euplotes woodruffi* | |
|---|---|---|---|---|---|---|
| | MAC[a,*] | MIC[b] | MAC[c] | MIC[†] | MAC[†] | MIC[†] |
| Genome size (Mbp) | 67.1 | 496 | 60.6 | 237 | 72.2 | 172 |
| N50 (bp) | 3745 | 27,807 | 3339 | 14,722 | 2702 | 44,656 |
| GC% | 31.36 | 28.44 | 37.05 | 32.17 | 36.56 | 35.31 |
| Number of contigs[‡] | 22,426 | 25,720 | 25,206 | 28,446 | 35,099 | 17,655 |
| Two-telomere contigs | 14,225 | - | 15,802 | - | 19,061 | - |
| Telomeric contigs | 20,336 | - | 21,165 | - | 28,294 | - |
| Single-gene telomeric contigs | 76.1% | - | 75.5% | - | 68.5% | - |
| Maximum number of genes on a telomeric contig | 8 | - | 7 | - | 8 | - |

a - **Swart et al., 2013**; b - **Chen et al., 2014**; c - **Chen et al., 2015**.

*This study used the MAC genome of *Oxytricha* from **Swart et al., 2013** instead of the long-read assembly in **Lindblad et al., 2019**, because the short MAC genomes in the present study were primarily assembled from Illumina reads, as in **Swart et al., 2013**. **Lindblad et al., 2019** updated **Swart et al., 2013** by including nanochromosomes captured in single long reads, which are currently not available for the other two species. The MIC genomes of *Tetmemena* and *E. woodruffi* were assembled to a similar N50 as the reference *O. trifallax* genome (**Chen et al., 2014**) for comparative analysis.

[†]Data from this study.

[‡]Telomere-bearing element (TBE) transposon contaminants in MAC contigs were removed (Methods). Therefore, 24 *Oxytricha* MAC contigs and 13 *Tetmemena* MAC contigs were removed from the published versions.

maps for the newly assembled genomes of *Tetmemena* and *E. woodruffi,* at a level comparable to the published reference for *O. trifallax* (**Chen et al., 2014**), which is appropriate for comparative analysis.

*Table 1* shows a comparison of genome features for the three species. The three MAC genomes are similar in size, with most nanochromosomes bearing only one gene. The size distributions of MAC chromosomes are similar for the three species, though slightly shorter for *E. woodruffi*, consistent with prior observation via gel electrophoresis (**Prescott, 1994**, *Figure 2—figure supplement 2*). Like *O. trifallax* (**Swart et al., 2013**), the maximum number of genes encoded on one chromosome is 7–8 (*Table 1*). Surprisingly, the MIC genome sizes differ substantially: the *Tetmemena* MIC genome assembly is 237 Mbp, nearly half that of *Oxytricha*. The *E. woodruffi* MIC genome assembly is even smaller, approximately 172 Mbp (*Table 1*).

The expansion of repetitive elements in the *Oxytricha* lineage may contribute to the difference in MIC genome sizes (*Figure 2A–C*). *Oxytricha* has a variety of tranposable elements (TEs) in the MIC, with telomere-bearing elements (TBEs) of the Tc1/*mariner* family the most abundant (**Chen et al., 2014**; **Chen and Landweber, 2016**, *Supplementary file 2*). A complete TBE transposon contains three open reading frames (ORFs). ORF1 encodes a 42kD transposase with a DDE-catalytic motif. Though present only in the germline, TBEs are so abundant in hypotrichs that some were partially recovered and assembled from whole cell DNA (**Chen and Landweber, 2016**). The *Oxytricha* MIC genome contains ~10,000 complete TBEs and ~24,000 partial TBEs, which occupy approximately 15.20% (75 Mbp) of the genome (*Figure 2A*, *Supplementary file 3*; **Chen et al., 2014**; **Chen and Landweber, 2016**). *Tetmemena*, on the other hand, has many fewer TBE ORFs and only 48 complete TBEs (*Supplementary file 3*), comprising 1.83% (4.3 Mbp) of its MIC genome (*Figure 2B*). *E. crassus* has also been reported to have an abundant transposon family called Tec elements (Transposon of <u>E</u>uplotes <u>c</u>rassus). Like TBEs, each Tec consists of three ORFs, and ORF1 also encodes a transposase from the Tc1/*mariner* family (**Baird et al., 1989**; **Krikau and Jahn, 1991**; **Jahn et al., 1993**; **Jahn et al., 1989**; **Klobutcher and Herrick, 1997**). The ~57 kD ORF2 encodes a tyrosine-type recombinase (**Doak et al., 2003**), and the 20kD ORF3 has unknown function (**Jahn et al., 1993**). Using the three

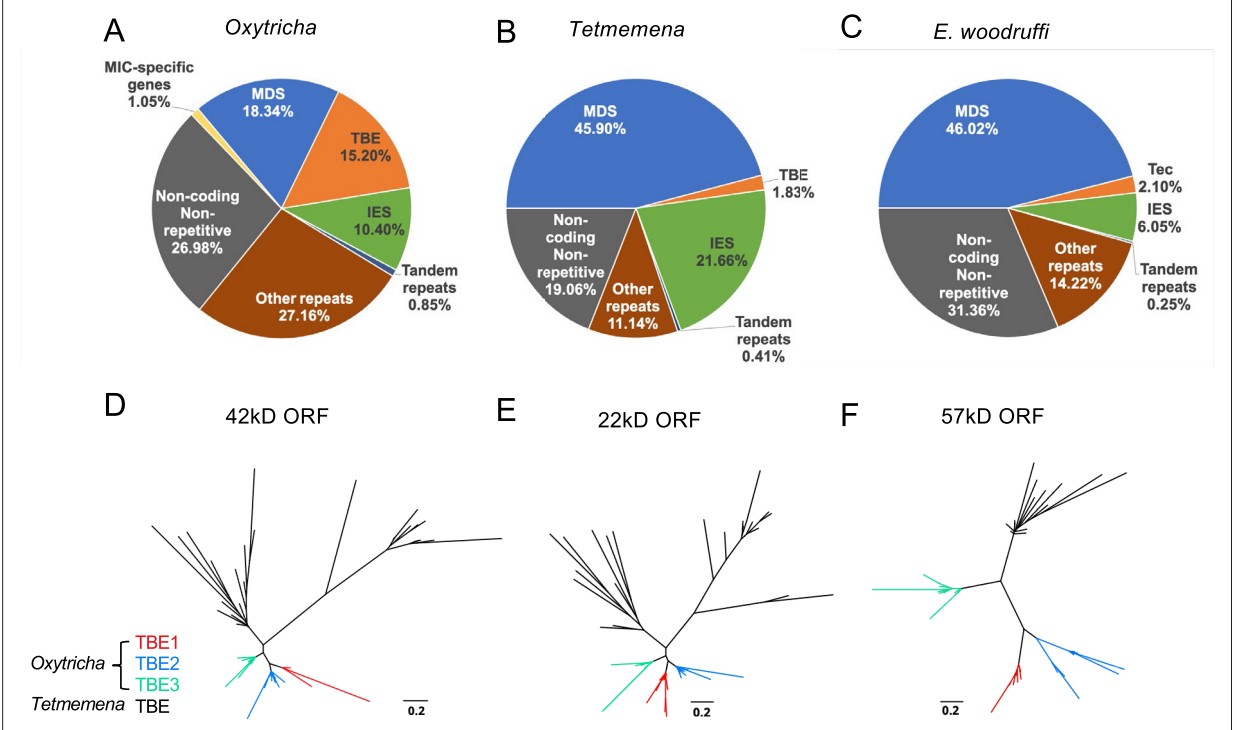

**Figure 2.** The three germline micronucleus (MIC) genomes differ in repeat content, especially transposable elements. (**A–C**) MIC genome categories for (**A**) *Oxytricha trifallax*, (**B**) *Tetmemena sp.*, and (**C**) *Euplotes woodruffi*. *Oxytricha* displays the greatest proportion of repetitive elements (telomere-bearing elements [TBE], other repeats, and tandem repeats) relative to the other species. *Oxytricha* MIC-specific genes were annotated in ***Chen et al., 2014***; ***Miller et al., 2021***. (**D–F**) Phylogenetic analysis of the three TBE open reading frames (ORFs) in *Oxytricha* and *Tetmemena*: (**D**) 42 kD, (**E**) 22 kD, and (**F**) 57 kD, suggest that TBE3 (green) is the ancestral transposon family in *Oxytricha*. For each ORF, 30 protein sequences from each species were randomly subsampled and maximum likelihood trees constructed using PhyML (***Guindon et al., 2010***).

The online version of this article includes the following figure supplement(s) for figure 2:

**Figure supplement 1.** Comparison of germline micronuclear (MIC) genome context of (**A**) Telomere-Bearing Element (TBE) and Transposon of *Euplotes crassus* (Tec) transposons and (**B**) other transposable elements in the three species.

**Figure supplement 2.** Length distribution of assembled somatic macronuclear (MAC) nanochromosomes in the three species.

ORFs of Tec1 and Tec2 as queries for search, we identified 74 complete Tec elements in *E. woodruffi*. Collectively, Tec ORFs occupy 3.6 Mbp, corresponding to only 2.1% of the MIC genome (***Figure 2C***). Notably, the transposase-encoding ORF1 is more abundant than the other two TBE/Tec ORFs in all three ciliates (***Supplementary file 3***), consistent with its proposed role in DNA cleavage during genome rearrangement in *Oxytricha* (***Nowacki et al., 2009***).

*Oxytricha* contains three families of TBEs. TBE3 appears to be the most ancient among hypotrichs, based on previous analysis of limited MIC genome data (***Chen and Landweber, 2016***). We constructed phylogenetic trees using randomly subsampled TBE sequences for all three ORFs from *Oxytricha* and *Tetmemena* (***Figure 2D–F***). This confirmed that only TBE3 is present in the *Tetmemena* MIC genome, as proposed in ***Chen and Landweber, 2016***. This also suggests that TBE1 and TBE2 expanded in *Oxytricha* after its divergence from other hypotrichous ciliates. As illustrated in ***Figure 2—figure supplement 1***, the MIC genome contexts of TBEs in *Oxytricha* and *Tetmemena* are similar, with many TE insertions within IESs, consistent with either IESs as hotspots for TE insertion or with the model (***Klobutcher and Herrick, 1997***) that some TE insertions may have generated IESs, as demonstrated in *Paramecium* (***Sellis et al., 2021***; ***Feng and Landweber, 2021***). Subsequent sequence evolution at the edges of IES/MDS pointers (***DuBois and Prescott, 1995***) can give rise to boundaries that no longer correspond precisely to TBE ends. For further discussion of the conservation of TBE locations, see the section, '*Oxytricha* and *Tetmemena* share conserved rearrangement junctions' below.

Additionally, Repeatmodeler/Repeatmasker identified that *Oxytricha* has more MIC repeats in the 'Other' category than *Tetmemena* or *E. woodruffi* (***Figure 2***, subcategories of repeat content in

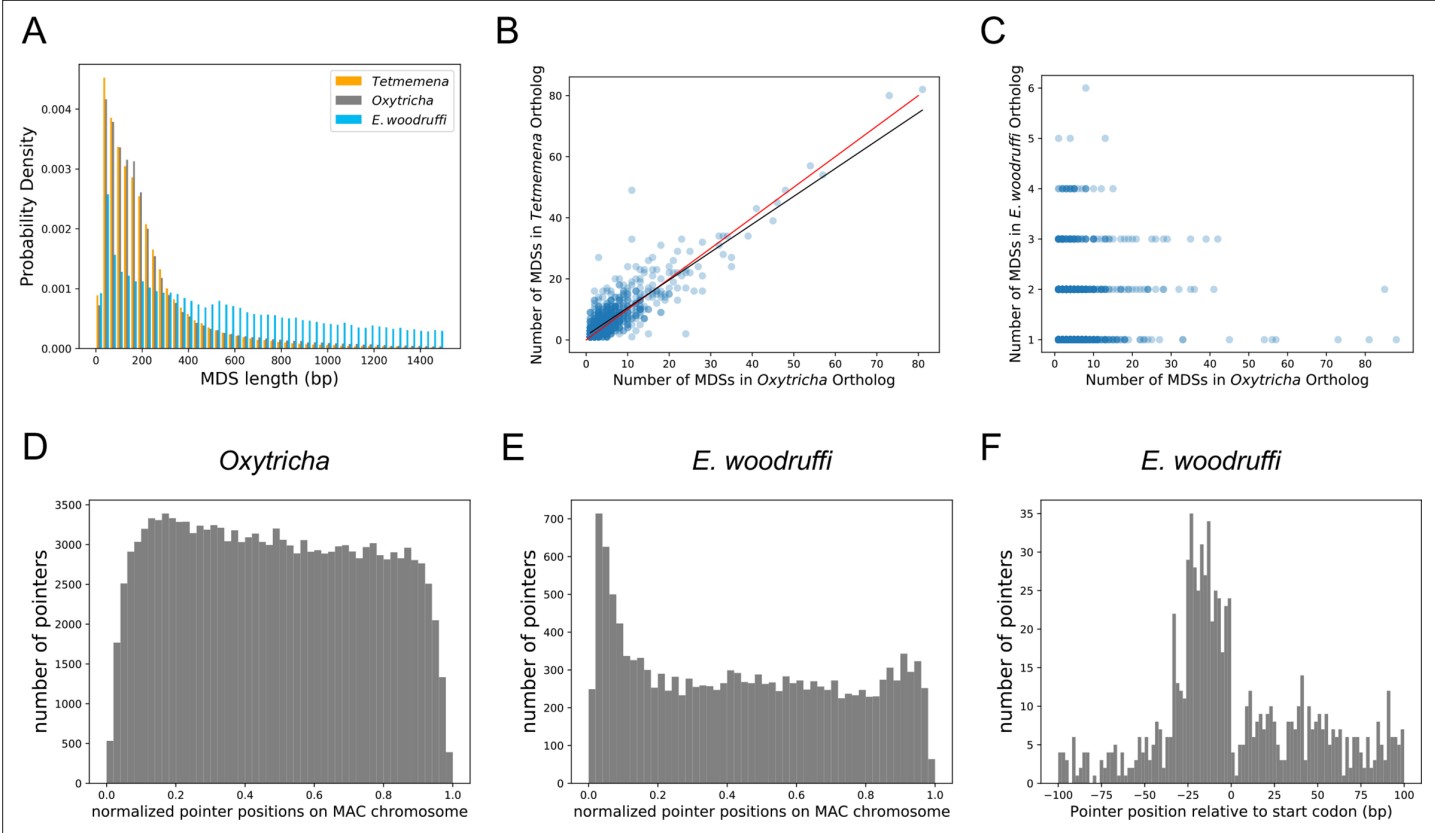

**Figure 3.** The three germline micronucleus genomes are interrupted by internally eliminated sequences (IESs) at different levels. (**A**) Macronuclear destined sequences (MDSs) of *Euplotes woodruffi* are longer compared to *Oxytricha* or *Tetmemena*. (**B**) Positive correlation between the numbers of MDSs for orthologous genes in *Tetmemena* and in *Oxytricha* for 903 single-gene orthologs. Black line is the function of linear regression ($R^2$=0.75). Red line is y=x. (**C**) Orthologs in *E. woodruffi* have fewer MDSs compared to *Oxytricha*, with no correlation ($R^2$=0.003). Note that many highly discontinuous genes in *Oxytricha* are IES-less in *E. woodruffi* (present on one MDS). 917 single-gene orthologs are shown. (**D**) Distribution of pointers on single-gene somatic macronucleus (MAC) chromosomes in *Oxytricha vs.* (**E**) *E. woodruffi*, with MAC chromosomes oriented in gene direction. Pointers significantly accumulate at the 5' end of single-gene MAC chromosomes in *E. woodruffi*. (**F**) Pointer positions on 3684 two-MDS MAC chromosomes demonstrate a preference upstream of the start codon.

The online version of this article includes the following figure supplement(s) for figure 3:

**Figure supplement 1.** Lengths of orthologs in *Oxytricha, Tetmemena* and *Euplotes woodruffi,* and the distribution of pointers on *Tetmemena* chromosomes.

**Figure supplement 2.** Scrambled and nonscrambled loci have distinct length distributions of internally eliminated sequences (IESs) and pointers.

*Supplementary file 2*). 214 Mbp of the *Oxytricha* MIC genome (43%, which is greater than 35.9% reported in *Chen et al., 2014* that used earlier versions of the software) is considered repetitive (including TBEs, tandem repeats, and other repeats in *Figure 2*), versus 31.7 Mbp for *Tetmemena* (13.4%) and 28.5 Mbp (16.8%) for *E. woodruffi*. *Oxytricha*'s additional ~180 Mbp in repeat content partially explains the significantly larger MIC genome size of *Oxytricha* versus the other spirotrich ciliates.

## The *E. woodruffi* genome has fewer IESs

We used the genome rearrangement annotation tool, Scrambled DNA Rearrangement Annotation Protocol (SDRAP, *Braun et al., 2022*) to annotate the MIC genomes of *Oxytricha, Tetmemena*, and *E. woodruffi* (Methods). Consistent with their close genetic distance, the genomes of *O. trifallax* and *Tetmemena* have similarly high levels of discontinuity (*Figure 3A*). We annotated over 215,299 MDSs in *Oxytricha* and over 215,624 in *Tetmemena* with similar MDS length distributions (*Figure 3A*). By contrast, *E. woodruffi* MDSs are typically longer, which indicates a less interrupted genome (*Figure 3A*). We compared the number of MDSs between single-copy orthologs for single-gene MAC

chromosomes across the three species and found that the orthologs have similar coding sequence (CDS) lengths (*Figure 3—figure supplement 1A–B*). There is a strong positive correlation between number of MDSs for orthologous genes in *Oxytricha* and *Tetmemena* (R²=0.75, *Figure 3B*). There is no correlation among number of MDSs between orthologs of *E. woodruffi* and *Oxytricha* (R²=0.003, *Figure 3C*), since *E. woodruffi* orthologs typically contain fewer MDSs.

The *E. woodruffi* genome is generally much less interrupted than that of *Oxytricha* or *Tetmemena*. 39.9% of MAC nanochromosomes in *E. woodruffi* lack IESs (IES-less nanochromosomes) compared to only 4.1 and 4.4% in *Oxytricha* and *Tetmemena*, respectively. The sparse IES distribution (as measured by plotting pointer distributions) in *E. woodruffi* displays a curious 5' end bias on single-gene MAC chromosomes, oriented in gene direction (*Figure 3E*). A weak 5' bias is also present in *Oxytricha* (*Figure 3D*) and *Tetmemena* (*Figure 3—figure supplement 1C*). In addition, *E. woodruffi* IESs preferentially accumulate in the 5' UTR, a short distance upstream of start codons (*Figure 3F*). Notably, the median distance between the 5' telomere addition site and the start codon in *E. woodruffi* is just 54 bp for single-gene chromosomes, approximately half that of *Oxytricha* (*Swart et al., 2013*).

## *E. woodruffi* has an intermediate level of genome scrambling

Scrambled genome rearrangements exist in all three species, which we report here for the first time in *Tetmemena* and the early diverged *E. woodruffi*. Previous studies have described scrambled genes with confirmed MIC-MAC rearrangement maps for a limited species of hypotrichs (*Chen et al., 2014*; *Chen et al., 2015*; *Hogan et al., 2001*; *Chang et al., 2005*; *Wong and Landweber, 2006*; *DuBois and Prescott, 1995*) and *Chilodonella* (*Katz and Kovner, 2010*; *Gao et al., 2014*) but not in *Euplotes*. Consistent with the phylogenetic placement of *Euplotes* as an earlier diverged outgroup to hypotrichs (*Lynn, 2008*; *Gao et al., 2016*), the *E. woodruffi* genome is scrambled, but it contains approximately half as many scrambled genes (2429 genes encoded on 1913 chromosomes, or 7.3% of genes), versus 15.6% scrambled in *O. trifallax* (3613 genes encoded on 2852 chromosomes) and 13.6% in *Tetmemena* (3371 genes encoded on 2556 chromosomes). The *E. woodruffi* lineage may therefore reflect an evolutionary intermediate stage between ancestral genomes with only modest levels of genome scrambling and the more massively scrambled genomes of hypotrichs.

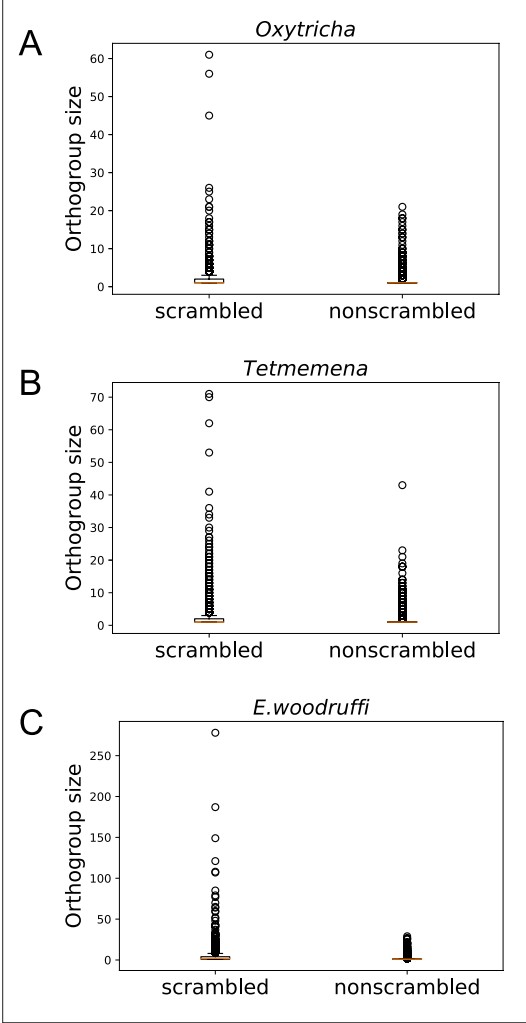

**Figure 4.** Scrambled genes have more paralogs than nonscrambled genes in the three species. Orthogroups containing at least one scrambled gene ('scrambled') are larger than orthogroups that lack scrambled genes ('nonscrambled') in (**A**) *Oxytricha*, (**B**) *Tetmemena,* and (**C**) *Euplotes woodruffi*.

The online version of this article includes the following figure supplement(s) for figure 4:

**Figure supplement 1.** An example of a *Euplotes woodruffi* scrambled gene locus containing paralogous macronuclear destined sequences (MDSs).

**Figure supplement 2.** The trend of scrambled loci to contain odd-even patterns may arise from partial duplication followed by mutation accumulation.

**Figure supplement 3.** Expression level of scrambled and nonscrambled genes in (**A**) *Oxytricha*, (**B**) *Tetmemena,* and (**C**) *Euplotes woodruffi*.

We infer that many genes were likely scrambled in the last common ancestor of *Oxytricha* and *Tetmemena*, because these two species share approximately half of their scrambled genes (*Supplementary file 4*). Furthermore, most scrambled genes are not new genes, since they possess at least one ortholog in other ciliate species (*Supplementary file 4*, *Supplementary file 5*).

## Scrambled genes are associated with local paralogy

Notably, scrambled genes in all three species generally have more paralogs (*Figure 4*). We identified orthogroups containing genes derived from the same gene in the last common ancestor of the three species (Methods). For each species, orthogroups with at least one scrambled gene are significantly larger than those containing no scrambled genes (p-value <1e−5, Mann-Whitney U test, *Figure 4A–C*). This association suggests a possible role of gene duplication in the origin of scrambled genes.

Scrambled pointers are generally longer than nonscrambled ones in all three species (*Figure 3—figure supplement 2*), consistent with prior observations (*Chen et al., 2014*) and the possibility that longer pointers participate in more complex rearrangements, including recombination between MDSs separated by greater distances (*Landweber et al., 2000*). Scrambled and nonscrambled IESs also differ in their length distribution (*Figure 3—figure supplement 2*). Curiously, scrambled 'pointers' in *E. woodruffi* can be as long as several hundred base pairs (median 48 bp, average 212 bp) unlike the more typical 2–20 bp canonical pointers. These long 'pointers' in *E. woodruffi* are more likely partial MDS duplications (*Figure 4—figure supplement 1A*). We also identified MDSs that map to two or more paralogous regions within the same MIC contig (*Supplementary file 6*), therefore representing MDS duplications and not alleles. Such paralogous regions could be alternatively incorporated into the rearranged MAC product. Moreover, we find that, for all three species, there are significantly more scrambled chromosomes than nonscrambled MAC chromosomes that contain at least one paralogous MDS (chi-square test, p-value <1e−10; *Supplementary file 6*). An example is shown in *Figure 4—figure supplement 1A* (MDS 7 and 7').

The presence of paralogous MDSs can contribute to the origin of scrambled rearrangements, as proposed in an elegant model by *Gao et al., 2015*; illustrated in *Figure 4—figure supplement 1B*. The model proposes that initial MDS duplications permit alternative use of either MDS copy into the mature MAC chromosome. As mutations accumulate in redundant paralogs, cells that incorporate the least decayed MDS regions into the MAC gene would have both a fitness advantage and a better match to the template RNA (*Nowacki et al., 2008*) that guides rearrangement, thus increasing the likelihood of incorporation into the MAC chromosome. The paralogous regions containing more mutations would gradually decay into IESs, and scrambled pointers eventually be reduced to a shorter length. The extended length 'pointers' that we identified in *E. woodruffi* may reflect an intermediate stage in the origin of scrambled genes (*Figure 4—figure supplement 1B*).

This model may generally explain the abundance and expansion of 'odd-even' patterns in ciliate scrambled genes (*Landweber et al., 2000*; *Burns et al., 2016*). As illustrated in *Figure 4—figure supplement 1A*, the even- and odd-numbered MDSs for many scrambled genes derive from different MIC genome clusters. The model predicts that the IES between MDS $n-1$ and $n+1$ often derives from ancestral duplication of a region containing MDS $n$ (*Figure 4—figure supplement 2A*). To test this hypothesis explicitly, we extracted from all odd-even scrambled loci in the three species all sets of corresponding MDS/IES pairs that are flanked by identical pointers on both sides, i.e., all pairs of scrambled MDSs and IESs, where the IES between MDS $n-1$ and $n+1$ is directly exchanged for MDS $n$ during DNA rearrangement (S1 and S2 in *Figure 4—figure supplement 2A*). To exclude the possibility of alleles confounding this analysis, MDS and IES pairs were only considered if they map to the same MIC contig. In *E. woodruffi*, the lengths of these MDS/IES pairs strongly correlate (Spearman correlation $\rho$ =0.755, p<1e−5, *Figure 4—figure supplement 2B*). Moreover, many MDS and IES sequence pairs also share sequence similarity, consistent with paralogy: for 248 MDS-IES pairs of similar length, 90.3% share a core sequence with ~97.5% identity across 8–100% of both the IES and MDS length. The lowest end of these observations is also compatible with an alternative model (*Chang et al., 2005*) in which direct recombination between IESs and MDSs at short repeats can lead to expansion of odd-even patterns. For *Oxytricha* and *Tetmemena*, the MDS and IES lengths for such MDS/IES pairs also display a weakly-positive correlation (p-values and Spearman correlation $\rho$ shown in *Figure 4—figure supplement 2D–E*). Remarkably, the odd-even-containing loci that are

species-specific, and therefore became scrambled more recently, have the strongest length correlation (*Figure 4—figure supplement 2C–E*) and more pairs that display sequence similarity (*Supplementary file 7*) relative to older loci (scrambled in two or more species). This result is consistent with an evolutionary process in which mutations accumulate in one copy of the MDS, gradually obscuring its sequence homology and ability to be incorporated as a functional MDS, and eventually its ability to be recognized by the template RNAs that guide DNA rearrangement. This analysis also suggests that most of the odd-even scrambled loci in *E. woodruffi* arose recently, because there is greater sequence similarity between MDSs and the corresponding IESs that they replace. Conversely, we infer that most loci that are scrambled in both *Oxytricha* and *Tetmemena* became scrambled earlier in evolution, since they display weaker sequence similarity between exchanged MDS and IES regions.

Scrambled and nonscrambled genes display nearly identical expression support (the presence of at least one read in all three replicates) in both *Oxytricha* (*Supplementary file 8*) and *Tetmemena. E. woodruffi* has slightly more expression support for nonscrambled vs. scrambled genes (*Figure 4—figure supplement 3*), which could be explained by more recent acquisition of thousands of scrambled loci in *E. woodruffi*. In some of those cases the nonscrambled paralogs may still contribute the major function. The distribution of expression levels is similar for scrambled vs. nonscrambled genes in all three species, supporting their authenticity (*Figure 4—figure supplement 3*), although in a Mann-Whitney U test, the average expression level of three replicates is significantly higher in nonscrambled genes for *Oxytricha* and *E. woodruffi*, but not significant for *Tetmemena*.

## *Oxytricha* and *Tetmemena* share conserved DNA rearrangement junctions

To understand the conservation of genome rearrangement patterns, we developed a pipeline guided by protein sequence alignment to compare pointer positions for orthologous genes between any two species (Methods, *Figure 5A*). We compared pointers for 2503 three-species single-copy orthologs. 4448 pointer locations are conserved between *Oxytricha* and *Tetmemena* on 1345 ortholog pairs (*Supplementary file 9*), representing 38.3% of pointers in these orthologs in *Oxytricha* and 30.9% in *Tetmemena*. For *Oxytricha/E. woodruffi* and *Tetmemena/E. woodruffi* comparisons, 56 and 58 pointer pairs are conserved, respectively. We also identified 23 pointer locations shared among all three species (*Supplementary file 9*, *Figure 5B*, *Figure 5—source data 1*).

To test if these pointer locations are genuinely conserved versus coincidental matching by chance, we performed a Monte Carlo simulation, as also used to study intron conservation (*Rogozin et al., 2003*). We randomly shuffled pointer positions on CDS regions 1000 times and counted the number of conserved pointer pairs expected for each simulation (Methods). Of the 1000 simulations, none exceeded the observed number of conserved pointer pairs between *Oxytricha* and *Tetmemena* (p-value <0.001), suggesting evolutionary conservation of pointer positions (*Supplementary file 9*). A similar result was obtained for pointers conserved in all three species (*Supplementary file 9*). However, the numbers of pointer pairs conserved between *Oxytricha/E. woodruffi* and *Tetmemena/E. woodruffi* is similar to the expectations by chance (*Supplementary file 9*). The low level of pointer conservation of either hypotrichs with *E. woodruffi* may reflect the smaller number of IESs in *E. woodruffi*; hence, most pointers would have arisen in the hypotrich lineage. Furthermore, *E. woodruffi* is genetically more distant from the two hypotrichs; hence, the accumulation of substitutions would obscure protein sequence homology, which we used to compare pointer locations. For ortholog pairs between *Oxytricha* and *Tetmemena*, scrambled pointers are significantly more conserved than nonscrambled ones (chi-square test, p-value <1e−10, *Supplementary file 10*). We also find that most pointer sequences differ even if the positions are conserved (*Figure 5B*, *Figure 5—source data 1*, *Supplementary file 11*), suggesting that substitutions may accumulate in pointers without substantially altering rearrangement boundaries.

*Oxytricha* and *Tetmemena* both contain a high copy number of TBE transposons (*Chen et al., 2014*; *Chen and Landweber, 2016*; *Supplementary file 3*). We investigated the level of TBE conservation between these two species. To identify orthologous insertions, we focus on TBE insertions in nonscrambled IESs on single-copy orthologs, which include 1706 *Oxytricha* TBEs inserted in 1296 nonscrambled IESs (multiple TBEs can be inserted into an IES) and 180 *Tetmemena* TBEs inserted into 170 nonscrambled IESs. We refer to the pointer flanking a TBE-containing IES as a *TBE pointer*. No TBE pointer locations are conserved between two species. This suggests that TBEs might invade the

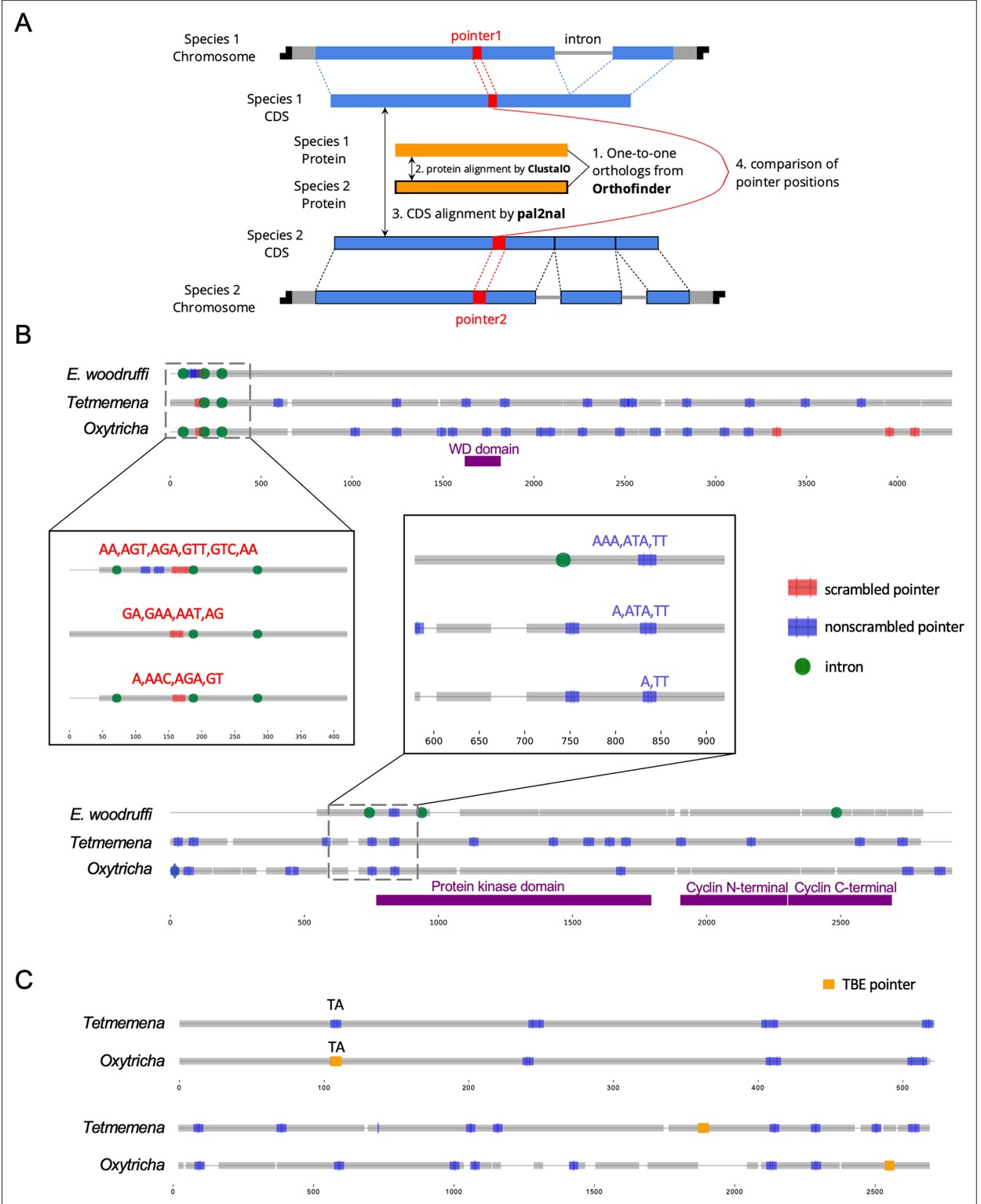

**Figure 5.** Identification and examples of conserved pointers. (**A**) Pipeline for comparison of pointer positions in orthologs. Orthologs are first grouped by OrthoFinder (*Emms and Kelly, 2019*), and protein sequences of single-copy orthologs aligned by Clustal Omega (*Sievers et al., 2011*). Then the protein alignments are reverse translated to coding sequence (CDS) alignments by a modified script of pal2nal (105, Methods). Pointers are annotated on the CDS alignments for comparison between any two orthologs. (**B**) Two examples of pointer conservation across three species. Gray lines represent the alignment of orthologous CDS regions, and boxes show magnified regions containing conserved pointers. The top panel shows a conserved scrambled pointer (*Oxytricha*: Contig889.1.g68; *Tetmemena*: LASU02015390.1.g1; *Euplotes woodruffi*: EUPWOO_MAC_30,105 .g1). The bottom panel

*Figure 5 continued on next page*

*Figure 5 continued*

shows a conserved nonscrambled pointer (*Oxytricha*: Contig19750.0.g98; *Tetmemena*: LASU02002033.1.g1; *E. woodruffi*: EUPWOO_MAC_31,621 .g1). Pointer sequences are noted, and commas indicate reading frame. Protein domains detected by HMMER (**Finn et al., 2011**) are marked in purple. (**C**) Examples of telomere-bearing element (TBE) insertions in nonscrambled internally eliminated sequences. The upper pair of sequences shows an *Oxytricha* TBE pointer (orange insertion of an incomplete TBE2 transposon containing the 42-kD and 57-kD open reading frames) conserved with a *Tetmemena* non-TBE pointer (*Oxytricha*: Contig736.1.g130; *Tetmemena*: LASU02012221.1.g1). Both species have a TA pointer at this junction. The bottom pair of sequences illustrates a case of nonconserved TBE pointers (*Oxytricha*: Contig17579.0.g71; *Tetmemena*: LASU02007616.1.g1).

The online version of this article includes the following source data and figure supplement(s) for figure 5:

**Source data 1.** Pointers conserved in all three species.

**Source data 2.** The telomere-bearing element (TBE) pointers in *Oxytricha* that are conserved with non-TBE pointers in *Tetmemena*.

**Figure supplement 1.** Examples of intron-internally eliminated sequence (IES) conversion across three species.

genomes of *Oxytricha* and *Tetmemena* independently, or still be actively mobile in the genome. Only 27 *Oxytricha* TBE pointers (containing 36 TBEs) are conserved with non-TBE pointers in *Tetmemena* (**Figure 5—source data 2**, **Figure 5C**). No *Tetmemena* TBE pointer is conserved with an *Oxytricha* non-TBE pointer. This suggests that TBE insertions may preferentially produce new rearrangement junctions instead of inserting into an existing IES.

## Intron locations sometimes coincide with DNA rearrangement junctions

Ciliate genomes are generally intron-poor. *Oxytricha* averages 1.7 introns/gene, *Tetmemena* has 1.1, and *E. woodruffi* has 2.2. Among three-species orthologs, intron locations sometimes map near pointer positions (within a 20-bp window, **Figure 5B**, **Figure 5—figure supplement 1**). IESs and introns are both noncoding regions that are removed from mature transcripts, though at different stages. A previous single-gene study observed that an IES in *Paraurostyla* overlaps the position of an intron in *Uroleptus*, *Urostyla*, and also the human homolog (**Chang et al., 2005**). This observation suggested an intron-IES conversion model in which the ability to eliminate non-CDS regions as either DNA or RNA provides a potential backup mechanism. Such interconversion has also been observed between two strains of *Stylonychia* (**Möllenbeck et al., 2006**). In the present study, we identified 174 potential cases of intron-IES conversion in the three species (**Figure 5—figure supplement 1**, **Supplementary file 12**): 103 (59.2%) *E. woodruffi* introns map near *Oxytricha*/*Tetmemena* pointers. We used a 20-bp window for this analysis, since one would only expect the boundaries of introns and IESs to coincide precisely if they were recent evolutionary conversions. A Monte Carlo simulation for these intron-IES comparisons (**Supplementary file 12**) revealed that p<0.001 for most three-species comparisons. For two-species comparisons, we identify 306 cases where an intron boundary in one species precisely coincides with a pointer sequence in another species, with strongest statistical support for the comparison between *Oxytricha* intron positions and *Tetmemena* IES junctions (p=0.008) (**Supplementary file 13**). Notably, *Tetmemena* intron locations rarely coincide with *Oxytricha* IESs (**Supplementary file 13**), suggesting a possible bias in the direction of intron-IES conversion during evolution.

The observation that *E. woodruffi* has the most introns but the smallest number of IESs per gene (**Figure 3**) is consistent with removal of intragenic non-CDS regions as either DNA or RNA. The intron-sparseness of ciliates is compatible with a hypothesis that it is advantageous to eliminate noncoding regions earlier at the DNA level, with intron deletion sometimes providing an opportunity for repair if they fail to be excised as IESs (**Chang et al., 2005**).

## Evolution of complex genome rearrangements: Russian doll genes

Genome rearrangements in the *Oxytricha* lineage can include overlapping and nested loci, with MDSs for different MAC loci embedded in each other (**Chen et al., 2014**; **Braun et al., 2018**). When multiple gene loci are nested in each other, these have been called Russian doll loci (**Braun et al., 2018**). *Oxytricha* contains two loci with five or more layers of nested genes (**Braun et al., 2018**). *Oxytricha* and *Tetmemena* display a high degree of synteny and conservation in both Russian doll loci. In the first Russian doll gene cluster, one nested gene (green) is present in *Oxytricha* but absent in *Tetmemena* (**Figure 6A**, **Figure 6—figure supplement 1**, **Figure 6—figure supplement 2**), confirmed by PCR (Methods). *Oxytricha* also has a complete TBE3 insertion in the green gene (**Figure 6A**, **Figure 6—figure supplement 1A**), hinting at a possible link between transposition and new gene insertion. In

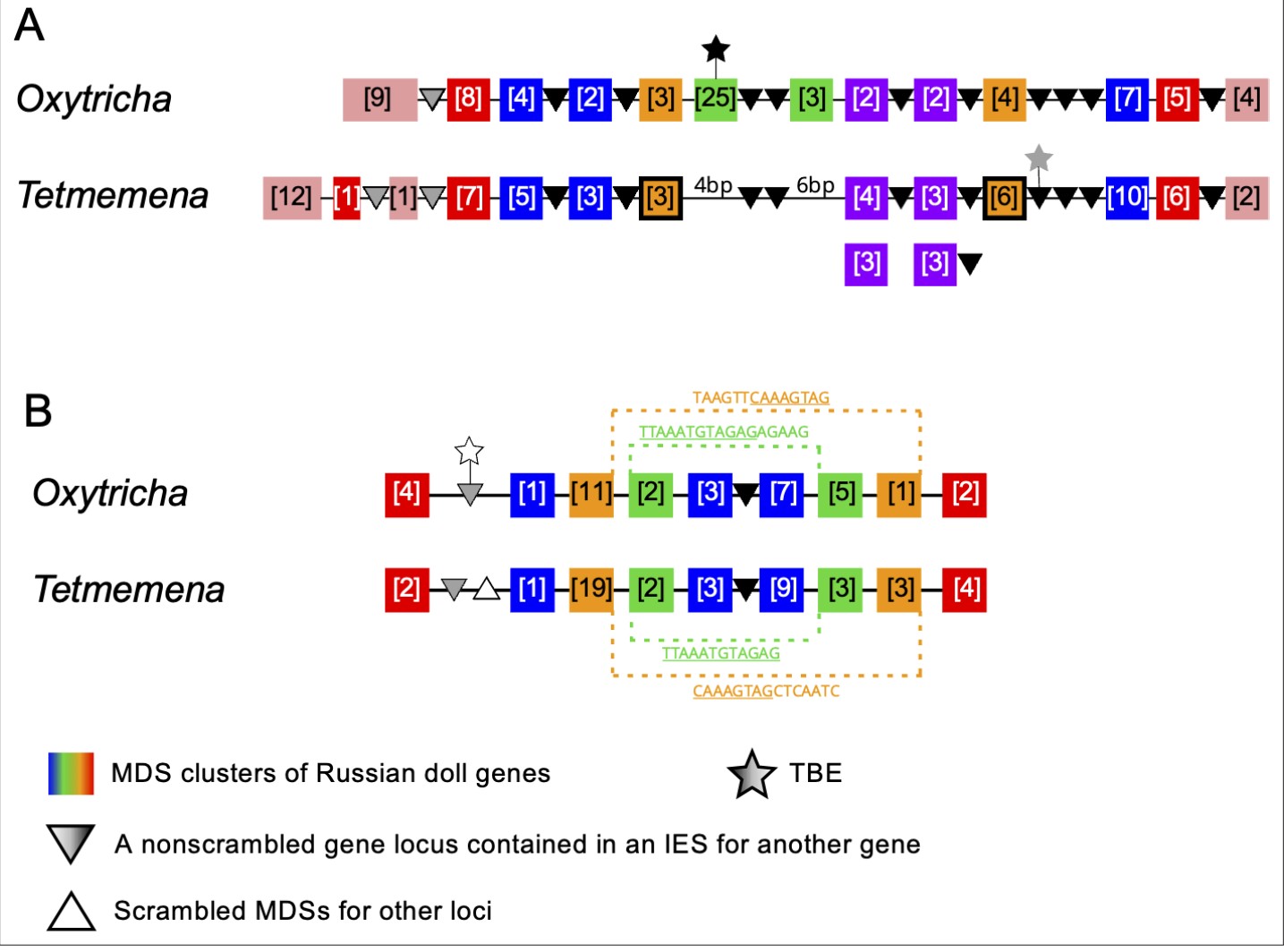

**Figure 6.** Synteny in 'Russian doll' loci in *Oxytricha* and *Tetmemena*. (**A**) Schematic comparison of the Russian doll gene cluster on *Oxytricha* germline micronucleus (MIC) contig OXYTRI_MIC_87484 vs. *Tetmemena* MIC contig TMEMEN_MIC_21461. Boxes of the same color represent clusters of macronuclear destined sequences (MDSs) for orthologous genes (detailed map in *Figure 6—figure supplement 1* and *Figure 6—figure supplement 2*). Numbers in brackets indicate the number of MDSs in each cluster, grouped by somatic macronucleus (MAC) chromosome. One nested gene (green) in *Oxytricha* is absent from *Tetmemena*. A two-gene chromosome (orange) that derives from seven MDSs in *Oxytricha* is processed as two single-gene chromosomes in *Tetmemena* instead (indicated by black border around orange boxes). The purple gene in *Oxytricha* has two paralogs in *Tetmemena*. Black triangles represent conserved, orthologous, and nonscrambled gene loci inserted between nested Russian doll genes. Empty triangle represents scrambled MDSs for other loci. Gray triangles, complete nonscrambled MAC loci embedded between gene layers in one species with no orthologous gene detected in the other species. Black star, a complete telomere-bearing element (TBE) transposon insertion. Gray star, a partial TBE insertion. (**B**) *Oxytricha* MIC contig OXYTRI_MIC_69233 vs. *Tetmemena* MIC contig TMEMEN_MIC_22886. Pointer sequences bridging the nested MDSs of orange and green genes are highlighted. The underlined pointer portions are conserved between species, e.g., the last 8 bp of the *Oxytricha* pointer, TAAGTTCAAAGTAG, is identical to the first 8 bp of CAAAGTAGCTCAATC in *Tetmemena*, illustrating pointer sliding (*DuBois and Prescott, 1995*), or gradual shifting of MDS/IES boundaries. White star indicates a decayed TBE with no open reading frame identified.

The online version of this article includes the following figure supplement(s) for figure 6:

**Figure supplement 1.** Detailed illustration of both Russian doll regions in *Figure 6*.

**Figure supplement 2.** Details of the Russian doll region in *Tetmemena* (TMEMEN_MIC_21461, *Figure 6A*).

addition, a two-gene chromosome in *Oxytricha* (orange) is present as two single-gene chromosomes in *Tetmemena* (*Figure 6A*, *Figure 6—figure supplement 1*). In *Oxytricha*, seven orange MDSs ligate across two other loci via an 18-bp pointer (TATATCTATACTAAACTT) to form a two-gene nanochromosome. However, in *Tetmemena*, telomeres are added to the ends of both gene loci instead, forming two independent MAC chromosomes (*Figure 6A*, *Figure 6—figure supplement 1*). The second

Russian doll locus has an example of a long, conserved pointer (orange dotted line) that bridges three other loci (the green and blue scrambled loci and one nonscrambled locus, *Figure 6B*). Close to this region is a decayed TBE insertion (769 bp) in *Oxytricha*. None of the *E. woodruffi* orthologs of both Russian doll loci maps to the same MIC contig, which suggests that the Russian doll clusters arose after the divergence of *Euplotes* from the common ancestor of *Oxytricha* and *Tetmemena*.

## Discussion

The highly diverse ciliate clade provides a valuable resource for evolutionary studies of genome rearrangement. However, full assembly and annotation of germline MIC genomes have concentrated on the model ciliates *Tetrahymena, Paramecium,* and *Oxytricha*. To provide insight into genome evolution in this lineage, we assembled and compared germline and somatic genomes of *Tetmemena sp.* and an outgroup, *E. woodruffi*, to that of *O. trifallax*. This expands our knowledge of the diversity of ciliate genome structures and the evolutionary origin of complex genome rearrangements.

Dramatic variation in transposon copy number (TBE and Tec elements) from the Tc1/*mariner* family appears to explain most of the variation in MIC genome size. In many eukaryotic taxa, genome size can differ dramatically even for closely related species, a phenomenon known as the 'C-value paradox' (*Thomas, 1971*). Our present observations are compatible with previous reports that the repeat content of the genome, especially transposon content, positively correlates with genome size (*Elliott and Gregory, 2015*).

*Oxytricha* has three TBE families in the MIC genome, but only TBE3 is present in *Tetmemena*, consistent with our previous conclusion that TBE3 is ancestral to the base of the transposon lineage in hypotrichous ciliates (*Chen and Landweber, 2016*). Tens of thousands of TBE1/2 transposons then expanded specifically in *Oxytricha*. Despite a high copy number of TBEs in both *Oxytricha* and *Tetmemena*, we find no identical TBE locations in nonscrambled IESs, even among syntenic Russian doll regions. These observations suggest that TBEs may be active in these genomes and contribute to the evolution of genome structure.

In the relatively IES-poor genome of *E. woodruffi*, IESs accumulate upstream of start codons, similar to the 5' bias of introns in intron-poor organisms (*Mourier and Jeffares, 2003*). The simplest model to explain 5' intron bias is homologous recombination between a reverse transcript of an intron-lacking mRNA and the original DNA locus to erase introns in the coding region (*Mourier and Jeffares, 2003*). A similar mechanism could simultaneously erase IESs in coding regions via germline recombination between the MIC chromosome and a reverse transcript; however, they are usually in different subcellular locations. More plausibly, a source for DNA recombination could be a MAC nanochromosome, since they are already abundant at high copy number, but another source could be by capture of a reverse transcript of a long non-coding template RNA that guides DNA rearrangement (*Nowacki et al., 2008*; *Lindblad et al., 2017*). Either recombination event in the germline would lead to loss of IESs, while retaining introns, but neither would necessarily provide a bias for IES-loss in coding regions. Any of these infrequent events would be meaningful on an evolutionary time scale, even if developmentally rare. The 5' bias of IESs could also reflect an evolutionary bias for continuous coding regions. Alternatively, upstream IESs might regulate gene expression or cell growth (*Sellis et al., 2021*), like some introns (*Parenteau et al., 2019*; *Morgan et al., 2019*).

This study investigated the evolution of scrambled genes by comparing *Oxytricha* and *Tetmemena* to *E. woodruffi*, as an earlier diverged representative of the spirotrich lineage. While *E. woodruffi* has approximately half as many scrambled genes as *Tetmemena* and *Oxytricha*, its genes are also much more continuous. For example, the most scrambled gene in *E. woodruffi,* encoding a DNA replication licensing factor (EUPWOO_MAC_28518, 3 kb), has only 20 scrambled junctions. The most scrambled gene in *Tetmemena* (LASU02015934.1, 14.7 kb, encoding a hydrocephalus-inducing-like protein) has 204 scrambled pointers, and the most scrambled gene in *Oxytricha* (Contig17454.0, 13.7 kb, encoding a dynein heavy chain family protein, *Chen et al., 2014*) is similarly complex, with 195 scrambled junctions. Together, these observations are consistent with our interpretation that *E. woodruffi* reflects an evolutionary intermediate stage, as it contains both fewer scrambled loci and fewer scrambled junctions within its scrambled loci. The observation that the most scrambled locus differs in each species is also consistent with the conclusion that complex gene architectures may continue to elaborate independently.

We observed that scrambled genes in each species tend to have more paralogs than nonscrambled genes. Similarly, in *C. uncinata* (*Maurer-Alcalá et al., 2018a*), a distantly related ciliate in the class *Phyllopharyngea* that also has scrambled genes, scrambled gene families (orthogroups) contain more genes (~2.9) than nonscrambled gene families (~1.3) (*Maurer-Alcalá et al., 2018a*). Apart from duplications at the gene level, *E. woodruffi* often contains partial MDS duplications at scrambled junctions, annotated as unusually long 'pointers' (*Figure 4—figure supplement 1*). We also demonstrate that odd-even scrambled patterns could readily arise from local duplications (*Figure 4—figure supplement 2*). These observations are most consistent with a simple model (*Gao et al., 2015*) in which local duplications permit combinatorial DNA recombination between paralogous germline regions, and mutation accumulation in either paralogs establishes an odd-even scrambled pattern that can propagate by weaving together segments from paralogous sources. Other proposed models include *Hoffman and Prescott, 1997* IES-invasion model that suggested that pairs of IESs could invade an MDS, and then subsequently recombine with another IES to yield odd-even scrambled regions; however, a previous examination did not find support for this model (*Chang et al., 2005*). *Prescott et al., 1998* also proposed that some odd-even scrambled loci could arise suddenly via reciprocal recombination with loops of A/T-rich DNA, but this does not exploit paralogy, only the high A/T content in the MIC. We previously proposed a gradual model (*Chang et al., 2005*; *Landweber et al., 2000*) in which MDS/IES recombination at short AT-rich repeats (precursors to pointers) could generate and propagate odd-even scrambled patterns. While limited comparisons of orthologs favored the stepwise recombination models (*Hogan et al., 2001*; *Chang et al., 2005*; *Wong and Landweber, 2006*; *DuBois and Prescott, 1995*), none of the earlier models accounted for the widespread existence of partial paralogy, revealed by genome assemblies.

Local duplications provide a buffer against mutations, allowing paralogous MDSs to repair the MAC locus during assembly of odd/even scrambled genes. Therefore, once an odd/even scrambled locus is established, a consequence is that evolution can only proceed in the direction of accumulating more scrambled junctions, as each new mutation in one paralog necessitates repair via incorporation of the other paralog (*Figure 4—figure supplement 1B*). This shortens the length of the respective MDSs and increases the number of recombination junctions, creating an evolutionary ratchet that drives the increase in scrambling. The lack of the presence of an error-free, continuous version of this locus in the germline reduces the possibility of losing the scrambled pattern from the MIC genome, relative to the trend toward decreasing MDS lengths as more mutations accumulate in either paralogs, with a resulting increase in the levels of scrambling and fragmentation (*Landweber, 2007*; *Speijer, 2008*). The only opportunity to repair a scrambled locus in the MIC would be a rare event that replaces the locus via recombination with a continuous version from the parental MAC, with the source being either parental MAC DNA or a reverse transcript of a template RNA (*Nowacki et al., 2008*; *Lindblad et al., 2017*), as discussed above.

Recent exciting reports have also described scrambled genomes in metazoa, including cephalopods (*Schmidbaur et al., 2022*; *Albertin et al., 2022*), but those events entail primarily evolutionary shuffling of gene order, without accompanying genome editing or repair. The ciliate lineage is remarkable in having evolved a sophisticated mechanism of RNA-guided genome editing that allows accurate and precise DNA repair of translocations and inversions. The future opportunity to harness this system to develop novel tools for genome editing outside of *Oxytricha* offers exciting directions.

## Methods

### DNA collection and sequencing of *Tetmemena sp.*

*Tetmemena sp.* (strain SeJ-2015; *Chen et al., 2015*) was isolated as a single cell from a stock culture and propagated as a clonal strain via vegetative (asexual) cell culture. Cells were cultured in Pringsheim media (0.11 mM $Na_2HPO_4$, 0.08 mM $MgSO_4$, 0.85 mM $Ca(NO_3)_2$, 0.35 mM KCl, pH 7.0) and fed with *Chlamydomonas reinhardtii*, together with 0.1%(v/v) of an overnight culture of non-virulent *Klebsiella pneumoniae*. Macronuclei and micronuclei were isolated using sucrose gradient centrifugation (*Lauth et al., 1976*). Genomic DNA was subsequently purified using the Nucleospin Tissue Kit (Takara Bio USA, Inc). Macronuclear DNA was sequenced and assembled in *Chen et al., 2015*. Micronuclear DNA was further size-selected via BluePippin (Sage Science) for PacBio sequencing, or via 0.6% (w/v)

SeaKem Gold agarose electrophoresis (Lonza) for Illumina sequencing. Micronuclear DNA purification and sequencing protocols are described in *Chen et al., 2014*.

## DNA collection and sequencing for *E. woodruffi*

*E. woodruffi* (strain Iz01) was cultured in Volvic water at room temperature and fed with green algae every 2–3 days. We fed cells with *C. reinhardtii* for MAC DNA collection, and switched to *Chlorogonium capillatum* for MIC DNA collection. In order to remove algal contamination, cells were starved for at least 2–3 days before collection. Cells were washed and concentrated as in *Chen et al., 2014*. Because MAC DNA is predominant in whole cell DNA, we used whole cell DNA (purified via NucleoSpin Tissue kit, Takara Bio USA, Inc) for MAC genome sequencing. Paired-end sequencing was performed on an Illumina Hiseq2000 at the Princeton University Genomics Core Facility.

MIC DNA was enriched from whole cell DNA and sequenced via three sequencing platforms (Illumina, Pacific Biosciences, and Oxford Nanopore Technologies). We used conventional and pulse-field gel electrophoresis (PFGE) to enrich MIC DNA:

1. High-molecular-weight DNA was separated from whole cell DNA by gel-electrophoresis (0.25% agarose gel at 4°C, 120 V for 4 hr). The top band was cut from the gel and purified with the QIAGEN QIAquick kit. The purified high-molecular-weight DNA was directly sent to the group of Dr. Robert Sebra at the Icahn School of Medicine at Mount Sinai for library construction and sequencing. BluePippin (Sage Science) separation was used before sequencing to select DNA >10 kb. DNA was sequenced on two platforms: Illumina HiSeq2500 (150 bp paired-end reads) and PacBio Sequel (SMRT reads).

2. High-molecular-weight DNA was also enriched by PFGE. *E. woodruffi* cells were mixed with 1% low-melt agarose to form plugs according to *Akematsu et al., 2017*, with addition of 1 hr incubation with 50 μg/ml RNase (Invitrogen AM2288) in 10 mM Tris-HCl (pH7.5) at 37°C for RNA depletion. After three washes of 1 hr with 1× TE buffer, the DNA plugs were incubated in 1 mM phenylmethylsulfonyl fluoride (PMSF) to inactivate proteinase K, followed by MspJI (New England Biolabs) digestion at $^mCNNR(9/13)$ sites to remove contaminant DNA ($^mC$ indicates C5-methylation or C5- hydroxymethylation). Previous reports have shown that no methylcytosine is detectable in vegetative cells of *Oxytricha* (*Bracht et al., 2012*), *Tetrahymena* (*Gorovsky et al., 1973*), and *Paramecium* (*Cummings et al., 1974*), suggesting that C5-methylation and C5-hydroxymethylation are rarely involved in the vegetative growth of the ciliate lineage. We also validated by qPCR that the quantity of two randomly selected MIC loci is not changed after the MspJI digestion. On the contrary, algal genomic DNA is significantly digested by MspJI. Based on these results, we conclude that MspJI digestion can be used to remove bacterial and algal DNA with C5-methylation and C5-hydroxymethylation, leaving *E. woodruffi* MIC DNA intact. The agarose plugs containing digested DNA were then inserted into wells of 1.0% Certified Megabase agarose gel (Bio-Rad) for PFGE (CHEF-DR II System, Bio-Rad). The DNA was separated at 6 V, 14°C with 0.5× TBE buffer at a 120° angle for 24 hr with switch time of 60–120 s. We validated by qPCR that the *E. woodruffi* MIC chromosomes were not mobilized from the well, while the MAC DNA migrated into the gel. The MIC DNA was then extracted by phenol-chloroform purification. Library preparation and sequencing were performed at Oxford Nanopore Technologies (New York, NY).

## MAC genome assembly of *E. woodruffi*

We assembled the MAC genome of *E. woodruffi* using the same pipeline for *Tetmemena sp.* (*Chen et al., 2015*) for comparative analysis: two draft genomes were assembled by SPAdes (*Bankevich et al., 2012*) and Trinity (*Grabherr et al., 2011*), and were then merged by CAP3 (*Huang and Madan, 1999*). Trinity, which is a software developed for de novo transcriptome assembly (*Grabherr et al., 2011*), has been used to assemble hypotrich MAC genomes (*Chen et al., 2015*) because their nano-chromosome genome structure is similar to transcriptomes, including properties such as variable copy number and alternative isoforms (*Lindblad et al., 2019*). Telomeric reads were mapped to contigs by BLAT (*Kent, 2002*), and contigs were further extended and capped by telomeres when at least five reads pile up at a position near ends by custom python scripts (https://github.com/yifeng-evo/Oxytricha_Tetmemena_Euplotes/tree/main/MAC_genome_telomere_capping) (*Feng, 2022a*). The mitochondrial DNA was removed if the contig has a TBLASTX (*Camacho et al., 2009*) hit on the *Oxytricha* mitochondrial genome (Genbank accession JN383842.1 and JN383843.1) or two *Euplotes* mitochondrial genomes (*Euplotes minuta* GQ903130.1, *E. crassus* GQ903131.1). Algal contigs were

removed by BLASTN to all *C. reinhardtii* nucleotide sequences downloaded from Genbank. Non-telomeric contigs were mapped to bacterial NR by BLASTX to remove bacterial contaminations. The genome was further compressed by CD-HIT (*Fu et al., 2012*) in two steps: (1) contigs <500 bp were removed if 90% of the short contig can be aligned to a contig ≥ 500 bp with 90% similarity (-c 0.9 -aS 0.9 -uS 0.1); (2) then the genome was compressed by 95% similarity (-c 0.95 -aS 0.9 -uS 0.1). Contigs shorter than 500 bp without telomeres were removed. Nine contigs, likely Tec contaminants from the MIC genome, were also excluded (Tblastn, '-db_gencode 10 -evalue 1e-5'), and they could be assembled due to the high copy number in the MIC genome (47, 48, Genbank accessions of Tec ORFs are AAA62601.1, AAA62602.1, AAA62603.1, AAA91339.1, AAA91340.1, AAA91341.1, AAA91342.1).

## RNA sequencing of *E. woodruffi* and *Tetmemena sp.*

Three biological replicates of total RNA was isolated from asexually growing *E. woodruffi* and *Tetmemena sp.* cells using TRIzol reagent (Thermo Fisher Scientific) and enriched for the poly(A)+fraction using the NEBNext Poly(A) mRNA Magnetic Isolation Module (New England Biolabs). Stranded RNA-seq libraries were constructed using the ScriptSeq v2 RNA-seq library preparation kit (Epicentre) and sequenced on an Illumina Nextseq500 at the Columbia Genome Center. For *E. woodruffi*, the transcriptome was assembled by Trinity (*Grabherr et al., 2011*), and transcript alignments to the MAC genome were generated by PASA (*Haas et al., 2003*).

## Gene prediction of the *E. woodruffi* MAC genome and validation of MAC genome completeness

We followed the gene prediction pipeline developed by the Broad institute (https://github.com/PASApipeline/PASApipeline/wiki); using EVidenceModeler (EVM, *Haas et al., 2008*) to generate the final gene predictions. EVM produced gene structures by weighted combination of evidence from three resources: *ab initio* prediction, protein alignments, and transcript alignments (the weight was 3, 3, and 10 respectively). *Ab initio* prediction was generated by BRAKER2 pipeline (*Brůna et al., 2021*). Protein alignments for EVM were generated by mapping *Oxytricha* proteins to the *E. woodruffi* MAC genome by Exonerate (*Slater and Birney, 2005*). EVM predicted 33,379 genes on MAC chromosomes with at least one telomere.

We assessed MAC genome completeness using three methods: (1) 28,294 (80.6%) of the 35,099 *E. woodruffi* MAC contigs have at least one telomere. (2) In the *E. woodruffi* genes predicted on telomeric contigs, 88.8% of BUSCO (*Simão et al., 2015*; *Manni et al., 2021*) genes in the lineage database alveolata_odb10 were identified as complete. Within the 171 BUSCO genes, 135 are complete and single-copy, 17 are complete and duplicated, 7 are fragmented, and 12 are missing. This represents the best *Euplotes* MAC genome assembly available. (3) We identified 51 tRNA genes encoding all 20 amino acids by tRNAscan-SE (*Lowe and Eddy, 1997*) in the MAC genome, including two suppressor tRNAs of UAA and UAG.

## MIC genome assembly of *Tetmemena sp.*

The MIC genome of *Tetmemena* was assembled with a hybrid approach to combine reads from different sequencing platforms. *Tetmemena* Illumina reads were first assembled by SPAdes (77, parameters '-k 21,33,55,77,99,127 –careful'). PacBio reads were error corrected by FMLRC (*Wang et al., 2018*) using Illumina reads with default parameters. Corrected PacBio reads were aligned to both the MAC genome and the Illumina MIC assembly with BLASTN. Reads were removed if they start or end with telomeres or are aligned better to the MAC. The remaining reads were assembled with wtdbg2 (*Ruan and Li, 2020*, parameters '-x rs'). The PacBio assembly was polished by Pilon (*Walker et al., 2014*) with the '--diploid' option. The Illumina and PacBio assemblies were merged by quick-merge (*Chakraborty et al., 2016*) with the '-l 5000' option.

## MIC genome assembly of *E. woodruffi*

The MIC genome of *E. woodruffi* was assembled using a similar procedure as described above for *Tetmemena*. *E. woodruffi* reads were filtered to remove bacterial contamination, including abundant high-GC-content contaminants, possibly endosymbionts (*Boscaro et al., 2019*). Nanopore reads with GC content ≥55% were assembled by Flye (*Kolmogorov et al., 2019*) with the parameter '--meta' for metagenomic assembly of bacterial contigs. We used kaiju (*Menzel et al., 2016*) to identify bacteria

taxa for these contigs. 9 of 10 top-covered contigs derive from Proteobacteria, from which many *Euplotes* symbionts derive (*Boscaro et al., 2019*). Bacterial contamination was removed from Illumina reads if perfectly mapping to these metagenomic contigs by Bowtie2 (*Langmead and Salzberg, 2012*). The cleaned Illumina reads were then assembled by SPAdes with '-k 21,33,55,77,99,127' (*Bankevich et al., 2012*). Pacbio raw reads and Nanopore raw reads with GC content <55% were aligned to a concatenated database containing both the MAC genome and the Illumina MIC assembly with BLASTN. Reads were removed if they start or end with telomeres or align better to the MAC. Remaining PacBio/Nanopore reads were assembled by Flye with '--meta' mode. The PacBio-Nanopore assembly was polished by Pilon with the '--diploid' option. Illumina and PacBio-Nanopore assemblies were merged by quickmerge with the '-l 10000' option. Contigs shorter than 1 kb were removed.

## MIC genome decontamination

The draft MIC genome of *Tetmemena* was first mapped to telomeric MAC contigs by BLASTN. MIC contigs containing MDSs were included in the final assembly. The rest of the MIC contigs were filtered by a decontamination pipeline: (1) contigs were aligned to the *K. pneumoniae* genome, *C. reinhardtii* genome, and the *Oxytricha* mitochondrial genome by BLASTN to remove contaminants; (2) the remaining contigs were then searched against the bacteria NR database and a ciliate protein database (including protein sequences annotated in *Tetrahymena thermophila*: http://www.ciliate.org/system/downloads/tet-latest/4-Protein%20fasta.fasta; *Paramecium tetraurelia*: http://paramecium.cgm.cnrs-gif.fr; and *O. trifallax*: https://oxy.ciliate.org) by BLASTX. Contigs with higher bit score to bacteria NR or G+C >45% were removed. The *E. woodruffi* MIC genome was decontaminated, similarly, with addition of all *Chlorogonium* sequences (the algal food source) on NCBI and the two *Euplotes* mitochondrial genomes (*E. minuta* GQ903130.1, *E. crassus* GQ903131.1) to filter contaminants.

## Repeat identification

The repeat content in the MIC genomes was identified by RepeatModeler 1.0.10 (*Smit and Hubley, 2008*) and RepeatMasker 4.0.7 (*Smit et al., 2013*) with default parameters.

## TBE/Tec detection

Representative *Oxytricha* TBE ORFs (Genbank accession AAB42034.1, AAB42016.1, and AAB42018.1) were used as queries to search TBEs in the *Oxytricha* and *Tetmemena* MIC genomes by TBLASTN (-db_gencode 6 -evalue 1e-7 -max_target_seqs 30000). Tec ORFs were similarly detected by using *E. crassus* Tec1 and Tec2 ORFs as queries (-db_gencode 10 -evalue 1e-5 -max_target_seqs 30000, Genbank accessions of Tec ORFs are AAA62601.1, AAA62602.1, AAA62603.1, AAA91339.1, AAA91340.1, AAA91341.1, AAA91342.1). Complete TBEs/Tecs were determined by custom python scripts when three ORFs are within 2000 bp from each other and in correct orientation (https://github.com/yifeng-evo/Oxytricha_Tetmemena_Euplotes/tree/main/TBE_ORFs/TBE_to_oxy_genome_tblastn_parse.py, *Chen and Landweber, 2016*). 30 TBE ORFs with >70% completeness were subsampled from each species for phylogenetic analysis (except for the 57 kD ORF in *Tetmemena*, for which 21 were subsampled). The subsampled TBE ORFs were aligned using MUSCLE (*Edgar, 2004*), and the alignments were trimmed by trimAl '-automated1' (*Capella-Gutiérrez et al., 2009*). Phylogenetic trees were constructed using PhyML 3.3 (*Guindon et al., 2010*).

## Rearrangement annotations

SDRAP (*Braun et al., 2022*) was used to annotate MDSs, pointers, and MIC-specific regions (minimum percent identity for preliminary match annotation = 95, minimum percent identity for additional match annotation = 90, minimum length of pointer annotation = 2). SDRAP requires MAC and MIC genomes as input. For the SDRAP annotation of *Oxytricha*, we used the MAC genome from *Swart et al., 2013* instead of the latest hybrid assembly that incorporated PacBio reads (*Lindblad et al., 2019*), because the former version was primarily based on Illumina reads, similar to the MAC genomes of *Tetmemena* (7, Genbank GCA_001273295.2) and *E. woodruffi* which are also Illumina assemblies. *Oxytricha* and *Tetmemena* MAC genomes were preprocessed by removing MAC contigs with TBE ORFs, considered MIC contaminants (*Chen and Landweber, 2016*). SDRAP is a new program that can output the rearrangement annotations with minor differences from *Chen et al., 2014*, but most annotations are robust (*Figure 3—figure supplement 2*). Scrambled and nonscrambled junctions/IESs were annotated

by custom python scripts (https://github.com/yifeng-evo/Oxytricha_Tetmemena_Euplotes/tree/main/scrambled_nonscrambled_IES_pointer).

## MIC genome categories

Each MIC genome region is assigned to only one category in *Figure 2A–C*, even if it belongs to more than one category. The assignment is based on the following priority: MDS, TBE/Tec, MIC genes (only available for *Oxytricha*, which has developmental RNA-seq data), IES, tandem repeats, other repeats, and non-coding non-repetitive regions. For example, an MIC region can be a TBE in an IES, and it is only considered as TBE in *Figure 2A–C*.

## Ortholog comparison pipeline and Monte Carlo simulations

Orthogroups of genes on telomeric MAC contigs were detected by OrthoFinder with '-S blast' (*Emms and Kelly, 2019*). Single-copy orthologs were aligned by Clustal Omega (*Sievers et al., 2011*). Protein alignments were reversely translated to CDS alignments by a modified script of pal2nal (*Suyama et al., 2006*, https://github.com/yifeng-evo/Oxytricha_Tetmemena_Euplotes/tree/main/Ortholog_comparison/pal2nal.pl). Two modifications were made in the script: (1) the modified script allows pal2nal to take different genetic codes for three sequences (-codontable 6,6,10); (2) the script also fixed an error in the original pal2nal script in which codontable 10 for the Euplotid nuclear code was the same as the universal code. Visualization of pointer positions and intron locations on orthologs was implemented by a custom python script (https://github.com/yifeng-evo/Oxytricha_Tetmemena_Euplotes/blob/main/Ortholog_comparison/visualization_of_ortholog_comparison.py). Pointer positions or intron locations are considered conserved if they are within a 20-bp alignment window on the CDS alignment. Protein domains were annotated by HMMER (*Finn et al., 2011*). We performed Monte Carlo simulations by randomly shuffling pointer locations on the CDS but keeping their original position distribution. This was implemented by a custom python script, which transforms the CDS to a circle, rotates pointer positions on the circle, and outputs the shuffled position on the re-linearized CDS (https://github.com/yifeng-evo/Oxytricha_Tetmemena_Euplotes/blob/main/Ortholog_comparison/shuffle_simulation.py). The null hypothesis of the Monte Carlo test is that pointer positions are conserved by chance. p-Value of Monte Carlo test is given by $N_{expected>observed}/N_{total}$ ($N_{expected>observed}$ is the number of simulations when there are more conserved pointers in the simulation than the observation from real data, $N_{total} = 1000$ in this study).

## PCR validation of Russian doll locus

The complex Russian doll locus on MIC contig TMEMEN_MIC_21461 in *Tetmemena* was validated by PCR to confirm the *Tetmemena* MIC genome assembly. *Tetmemena* micronuclear DNA was purified as described previously and used as template for PCR using PrimeSTAR Max DNA polymerase (Takara Bio). 11 primer sets (*Supplementary file 14*) were designed to amplify products between 3 kb and 6 kb in length, with overlapping regions between consecutive primer pairs. The resulting PCR products were visualized through agarose gel electrophoresis, and bands of the expected size were extracted using a Monarch DNA Gel Extraction Kit (New England Biolabs). The purified gel bands were cloned using a TOPO XL-2 Complete PCR Cloning Kit (Invitrogen), transformed into One Shot OmniMAX 2 T1R *E. coli* cells (Invitrogen), and individual clones were grown and their plasmids harvested with a QIAprep Spin Miniprep Kit (QIAGEN). The plasmid ends were Sanger sequenced, as well as the region where the *Oxytricha* MIC assembly contains inserted MDSs (Genewiz). Sanger sequencing reads were mapped to the *Tetmemena* MIC contig TMEMEN_MIC_21461 and visualized using Geneious Prime 2021.1.1 (https://www.geneious.com).

## Availability of data and materials

Custom scripts are public on https://github.com/yifeng-evo/Oxytricha_Tetmemena_Euplotes, (*Feng, 2022b* copy archived at swh:1:rev:fd66a0efeaf9feb2d79e183313192d641b4e5400). DNA-seq reads and genome assemblies are available at GenBank under Bioprojects PRJNA694964 (*Tetmemena sp.*) and PRJNA781979 (*E. woodruffi*). Genbank accession numbers for genomes are JAJKFJ000000000 (*Tetmemena sp.* Micronucleus genome), JAJLLS000000000 (*E. woodruffi* Micronucleus genome), and JAJLLT000000000 (*E. woodruffi* Macronucleus genome).

Three replicates of RNA-seq reads for vegetative cells are available at GenBank under accession numbers of SRR21815378, SRR21815379, and SRR21815380 for *E. woodruffi* and SRR21817702, SRR21817703, and SRR21817704 for *Tetmemena sp.*

MDS annotations for three species are available at https://doi.org/10.5061/dryad.5dv41ns96 and https://knot.math.usf.edu/mds_ies_db/2022/downloads.html (please select species from the drop-down menu).

## Acknowledgements

We thank Toshinobu Suzaki (Kobe University) for the gift of *E. woodruffi* (strain Iz01 from Shizuoka Prefecture) and *Chlorogonium capillatum*. We thank Sheela George for laboratory support and help with cell collection. We thank David Dai, Eoghan Harrington, John Beaulaurier, and Sissel Juul at Oxford Nanopore Technologies in New York for providing sequencing and advice. We thank Robert Sebra and Melissa Smith for advice and PacBio sequencing. We thank Takahiko Akematsu, Lorraine Symington, and Lea Marie for helping with PFGE. We thank Kaiyi Zhu, Shaojie He, Molly Przeworski, Harmen Bussemaker, and Nataša Jonoska for advice on Monte Carlo simulations. We also thank Scott Roy, Samuel Sternberg, Bill Jack, and all current and past Landweber lab members for discussion about the origin of scrambled genes, as well as David Prescott and Klaus Heckmann for inspiration, and Margarita T Angelova, Sindhuja Devanapally, Danylo Villano and Kehan Bao for comments on the manuscript. This work was supported by the National Institutes of Health, R35GM122555, and National Science Foundation, DMS1764366, and the National Center for Genome Analysis Support computing resources (supported by National Science Foundation DBI1062432, ABI1458641, and ABI1759906 to Indiana University). Rafik Neme was supported by the Pew Latin American Fellows Program.

## Additional information

### Competing interests

Leslie Y Beh: The author is currently employed by Illumina. Xiao Chen: employed by Pacific Biosciences. The other authors declare that no competing interests exist.

### Funding

| Funder | Grant reference number | Author |
| --- | --- | --- |
| National Institutes of Health | R35GM122555 | Laura F Landweber<br>Yi Feng<br>Leslie Y Beh<br>Michael W Lu |
| National Science Foundation | DMS1764366 | Yi Feng |
| Pew Latin American Fellows Program | | Rafik Neme |
| National Science Foundation | DBI1062432 | Yi Feng |
| National Science Foundation | ABI1458641 | Yi Feng |
| National Science Foundation | ABI1759906 | Yi Feng |

The funders had no role in study design, data collection and interpretation, or the decision to submit the work for publication.

### Author contributions

Yi Feng, Conceptualization, Data curation, Formal analysis, Investigation, Visualization, Methodology, Writing - original draft, Writing – review and editing; Rafik Neme, Conceptualization, Methodology; Leslie Y Beh, Xiao Chen, Investigation; Jasper Braun, Software; Michael W Lu, Investigation, Validation;

Laura F Landweber, Conceptualization, Supervision, Funding acquisition, Investigation, Project administration, Writing – review and editing

**Author ORCIDs**
Yi Feng http://orcid.org/0000-0002-2393-1700
Rafik Neme http://orcid.org/0000-0001-8462-5291
Xiao Chen http://orcid.org/0000-0002-1432-268X
Jasper Braun http://orcid.org/0000-0003-1250-4399
Michael W Lu http://orcid.org/0000-0002-4926-8839
Laura F Landweber http://orcid.org/0000-0002-7030-8540

**Decision letter and Author response**
Decision letter https://doi.org/10.7554/eLife.82979.sa1
Author response https://doi.org/10.7554/eLife.82979.sa2

## Additional files

**Supplementary files**
• Supplementary file 1. Sequencing depth statistics for germline micronucleus (MIC) genome assemblies. *Sequencing data from *Chen et al., 2014*. **Raw reads were mapped to the MIC genome assembly by Minimap2 and Bowtie2 (*Langmead and Salzberg, 2012*). Average coverage was calculated with BBmap (sourceforge.net/projects/bbmap/) pileup.sh for macronuclear destined sequence-containing contigs in the MIC genome assembly.

• Supplementary file 2. Subcategories of repeat content in the three species. Repeat content of the three genomes, as annotated by Repeatmasker (*Smit et al., 2013*) with additional manual annotation of Telomere-Bearing Element (TBE)/Transposon of *Euplotes crassus* (TEC) elements. The numbers may differ from *Figure 2A–C* because some repeats are assigned as other germline micronucleus (MIC) categories in the pie charts (Methods). For example, a MIC region which is both an internally eliminated sequence (IES) and satellite, is assigned as IES in *Figure 2A–C*, but is counted as a satellite in this table.

• Supplementary file 3. Telomere-bearing elements (TBE)/transposon of *Euplotes crassus* (TEC) elements open reading frames in three species. * Differs from 10,109 in Chen et al. (*Chen and Landweber, 2016*) because we used different versions of BLAST and custom python scripts to identify complete TBEs (see Methods).

• Supplementary file 4. Orthology among scrambled and nonscrambled genes in the three species. * Ciliate database is generated by extracting all protein sequences in phylum Ciliophora (taxid: 5878) from NR database.

• Supplementary file 5. Summary of orthologs in each pair of species. The (*i,j*) cell shows the number of genes in species *i* with an ortholog in species *j*. * Genes with no ortholog detected by OrthoFinder (*Emms and Kelly, 2019*) in the other two species.

• Supplementary file 6. More scrambled somatic macronucleus (MAC) contigs contain at least one paralogous macronuclear destined sequence that may be involved in alternative rearrangement.

• Supplementary file 7. Macronuclear destined sequence (MDS)-internally eliminated sequence (IES) pairs share homologous sequences in the three species (related to *Figure 4—figure supplement 2*).

• Supplementary file 8. Genes with expression support in the three species.

• Supplementary file 9. Presence of conserved pointers in three species, with Monte Carlo simulations.

• Supplementary file 10. Scrambled pointers are more conserved than nonscrambled pointers.

• Supplementary file 11. Most pointers conserved in position are different in sequence.

• Supplementary file 12. Intron-IES conversion comparison in three species and Monte Carlo simulations.

• Supplementary file 13. Pairwise intron-IES conversion comparisons and Monte Carlo simulations.

• Supplementary file 14. PCR primers for validation of the Russian doll region in *Tetmemena* DNA (*Figure 6A*).

• MDAR checklist

## Data availability

Custom scripts are public on https://github.com/yifeng-evo/Oxytricha_Tetmemena_Euplotes, (copy archived at swh:1:rev:fd66a0efeaf9feb2d79e183313192d641b4e5400). DNA-seq reads and genome assemblies are available at GenBank under Bioprojects PRJNA694964 (*Tetmemena sp.*) and PRJNA781979 (*Euplotes woodruffi*). Genbank accession numbers for genomes are JAJKFJ000000000 (*Tetmemena sp.* Micronucleus genome), JAJLLS000000000 (*Euplotes woodruffi* Micronucleus genome), and JAJLLT000000000 (*Euplotes woodruffi* Macronucleus genome). Three replicates of RNA-seq reads for vegetative cells are available at GenBank under accession numbers of SRR21815378, SRR21815379, SRR21815380 for *E. woodruffi* and SRR21817702, SRR21817703 and SRR21817704 for *Tetmemena sp*. MDS annotations for three species are available at https://doi.org/10.5061/dryad.5dv41ns96 and https://knot.math.usf.edu/mds_ies_db/2022/downloads.html (please select species from the drop-down menu).

The following datasets were generated:

| Author(s) | Year | Dataset title | Dataset URL | Database and Identifier |
|---|---|---|---|---|
| Feng Y, Neme R, Beh LY, Chen X, Braun J, Lu MW, Landweber LF | 2022 | Euplotes woodruffi genome sequencing and assembly | https://www.ncbi.nlm.nih.gov/bioproject/PRJNA781979 | NCBI BioProject, PRJNA781979 |
| Feng Y, Neme R, Beh LY, Chen X, Braun J, Lu MW, Landweber LF | 2022 | Tetmemena sp. micronucleus genome sequencing and assembly | https://www.ncbi.nlm.nih.gov/bioproject/PRJNA694964 | NCBI BioProject, PRJNA694964 |
| Feng Y, Neme R, Beh LY, Chen X, Braun J, Lu MW, Landweber LF | 2022 | Euplotes woodruffi strain:lz01 | https://www.ncbi.nlm.nih.gov/bioproject/PRJNA781602 | NCBI BioProject, PRJNA781602 |
| Feng Y, Neme R, Beh LY, Chen X, Braun J, Lu MW, Landweber LF | 2022 | RNA-seq of Tetmemena sp | https://www.ncbi.nlm.nih.gov/bioproject/PRJNA887426 | NCBI BioProject, PRJNA887426 |
| Channagiri T, Braun J, Feng Y, Landweber LF | 2022 | MDS-IES database | https://knot.math.usf.edu/mds_ies_db/2022/downloads.html | MDSIESDB, db/2022/downloads |
| Feng Y, Neme R, Beh L, Chen X, Braun J, Lu M, Landweber L | 2022 | MDS and IES annotations for Euplotes woodruff, Tetmemena sp. and Oxytricha trifallax | https://doi.org/10.5061/dryad.5dv41ns96 | Dryad Digital Repository, 10.5061/dryad.5dv41ns96 |
| Feng Y, Neme R, Beh LY, Chen X, Braun J, MW Lu, Landweber LF | 2022 | Tetmemena sp. Micronucleus genome | https://www.ncbi.nlm.nih.gov/nuccore/JAJKFJ000000000 | NCBI GenBank, JAJKFJ000000000 |
| Feng Y, Neme R, Beh LY, Chen X, Braun J, MW Lu, Landweber LF | 2022 | Euplotes woodruffi Micronucleus genome | https://www.ncbi.nlm.nih.gov/nuccore/JAJLLS000000000 | NCBI GenBank, JAJLLS000000000 |
| Feng Y, Neme R, Beh LY, Chen X, Braun J, MW Lu, Landweber LF | 2022 | Euplotes woodruffi Macronucleus genome | https://www.ncbi.nlm.nih.gov/nuccore/JAJLLT000000000 | NCBI GenBank, JAJLLT000000000 |

The following previously published datasets were used:

| Author(s) | Year | Dataset title | Dataset URL | Database and Identifier |
|---|---|---|---|---|
| Chen X, Bracht JR, Goldman AD, Dolzhenko E, Clay DM, Swart EC, Perlman DH, Doak TG, Stuart A, Amemiya CT, Sebra RP, Landweber LF | 2014 | Oxytricha trifallax micronucleus genome | https://www.ncbi.nlm.nih.gov/assembly/GCA_000711775.1 | NCBI Assembly, GCA_000711775.1 |
| Swart EC, Bracht JR, Magrini V, Minx P, Chen X, Zhou Y, Khurana JS, Goldman AD, Nowacki M, Schotanus K, Jung S, Ly A, McGrath S, Haub K, Wiggins JL, Storton D, Matese JC, Parsons L, Chang WJ, Bowen MS, Stover NA, Jones TA, Eddy SR, Herrick GA, Doak TG, Wilson RK, Mardis ER, Landweber LF | 2013 | Oxytricha trifallax macronucleus genome | https://www.ncbi.nlm.nih.gov/assembly/GCA_000295675.1/ | NCBI Assembly, GCA_000295675.1 |
| Chen X, Jung S, Beh LY, Eddy SR, Landweber LF | 2015 | Tetmemena sp. macronucleus genome | https://www.ncbi.nlm.nih.gov/assembly/GCA_001273295.2 | NCBI Assembly, GCA_001273295.2 |
| Beh LY, Debelouchina GT, Clay DM, Thompson RE, Lindblad KA, Hutton ER, Bracht JR, Sebra RP, Muir TW, Landweber LF | 2019 | Genome-wide analysis of chromatin and transcription in the ciliates Oxytricha trifallax and Tetrahymena thermophila | https://www.ncbi.nlm.nih.gov/geo/query/acc.cgi?acc=GSE94421 | NCBI Gene Expression Omnibus, GSE94421 |

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
