## [Editor Report]

The study marks a significant advance in the field of evolutionary genomics of ciliates, an ancient and highly diverse eukaryotic phylum with many idiosyncrasies that teach us valuable lessons, inter alia, about sex and the plasticity of genomes. By focusing on two species from the same family, plus a more distant outgroup within the same class, this valuable study provides new and compelling information on evolutionary trends of genome rearrangement among different species of this interesting group of organisms. The work will be of interest to anyone interested in genome dynamics.

---

## [Decision Letter]

[Editors' note: this paper was reviewed by Review Commons.]

Thank you for submitting your article "Comparative genomics reveals insight into the evolutionary origin of massively scrambled genomes." for consideration by eLife. Your article has been reviewed by 2 peer reviewers at Review Commons, and the evaluation at eLife has been overseen by Detlef Weigel as Deputy Editor in discussion with several Senior and Reviewing Editors and an outside expert.

Based on your manuscript, the reviews and your responses, we invite you to submit a revised version incorporating the revisions as outlined in your response to the reviews.

When preparing your revisions, please also address the following points:

A major concern is that the genome assemblies are poor. Much of this is probably because of the technical challenges associated with this system, but BUSCO scores of 76% are and N50 <30kb for the MICs are red flags. This does make one wonder how confident one can be that some of the conclusions, such as those pertaining to paralogs, are not at least in part due to assembly artifacts. It is striking that most genes undergoing unscrambling have no orthologs in others species, and that is thus unclear what these genes actually do. Please provide additional evidence that the conclusions are unaffected by the limitations of the assemblies. Please also provide more statistics regarding the assemblies (e.g., expected lengths distribution of nanochromosomes in MACs, number of chromosomes in MICs vs contigs, etc.), and annotations (gene families -- how similar are they between species, how many "species-specific" genes [which the genomics community considers to be largely, though not entirely artifactual] etc.).

Please spell out more clearly what you consider the evolutionary forces that have increased scrambling. My reading is that you consider many of the scrambled genes not to be functionally very important, which would be in line with there being so few orthologs in other species. To this end, please provide some basic information as to how many of the scrambled genes have expression support, what the distribution of expression quantiles is etc. -- this could help to shore up these inferences.

Finally, the manuscript is overly long, and despite -- or perhaps because -- of this, it is not presented in the most accessible manner. For example, the abstract still struggles to explain the importance or relevance of the work. e.g. the last sentence is "Scrambled loci are more often associated with local duplications, supporting a simple model for the origin of scrambled genes via DNA duplication and decay". This would need some motivation, i.e., why should we care about scrambled genes per se? Similarly, why does the Introduction have an almost 1-page summary of the results?

---

## [Author Response]

Reviewer #1 (Evidence, reproducibility and clarity (Required)):Summary:Ciliates extensively rearrange their somatic genome every time a new somatic nucleus develops from the zygotic germline nucleus. In this manuscript, Feng et al. report the sequencing, assembly and annotation of the germline and somatic genomes of Euplotes woodruffi and the germline genome of Tetmemena sp. (whose somatic genome was sequenced and assembled by the same lab in 2015). They present a comparative analysis of developmentally programmed genome rearrangements in these two species and in the model ciliate Oxytricha trifallax. Their major findings are that:(i) E. woodruffi and Tetmemena sp. eliminate a smaller fraction of their germline genome (~54%) from their somatic macronucleus (MAC) than O. trifallax (>80%)(ii) Transposable elements (TE) represent a smaller fraction of the germline genome (~2%) in the first two ciliates than in O. trifallax (~15%). TEs are mainly located at the boundaries of germline chromosomes and in intergenic regions, but can also be found inside IESs(iii) Several thousands of genes are scrambled in the germline genome of all three speciesThe authors have also addressed the possible origin of gene scrambling. They report an interesting association with local paralogy and propose a model for the emergence of the odd-even pattern of gene unscrambling between two paralogous copies.Major comments:1. Based on the statistics presented in Table 1, genome assemblies are of good quality, with a reasonable N50 size of germline (MIC) contigs. It seems, however, that no entire MIC chromosome could be assembled, since no two-telomere contig is mentioned in the list. As proposed by the authors (p.7) the presence of numerous TEs at the boundaries of MIC contigs (Figure S1) may have hindered the assembly of MIC chromosome ends. I would have appreciated to have more information on the "other repeats" (which seem to differ from tandem repeats according to Figure 2) and their location along MIC contigs.

Subcategories of “other repeats” were included in Table S2 based on Repeatmasker annotations. We now analyzed the locations of other repeats in MIC contigs and include those as well in new Figure S1B. About 30% of “other” transposable elements are present at the boundaries of MIC contigs, which may also hinder the assembly. Notably, 35-45% of “other TEs” are in assembled, intergenic regions.

2. The definition of "Internal Eliminated Sequences" (IES) is not clear. The authors make a distinction between IESs and TEs. I understand that IESs are DNA segments that separate two macronuclear-destined sequences (MDS) in the germline genome. Thus they appear to be restricted to those regions that eventually yield gene-sized MAC chromosomes. IESs are eliminated between two pointers that may not be identical on both sides in case of scrambled genes. Some clarification is needed here.To illustrate my point: I found the statement "with many TE insertions within IESs, suggesting that TE insertions may have generated IESs" particularly confusing (p. 9 lines 5-6). Does this mean that IESs extend beyond the ends of inserted TEs? The legend of Figure S1 should also be clarified.

We clarified the text and legend. IESs can extend beyond the ends of inserted TEs, even if the original IES is a decayed TE, due to subsequent sequence evolution at the boundaries or if the original insertion was into an existing IES. David Prescott referred to sequence evolution at the edges of IESs as “pointer sliding” (ref.36).

3. p. 10 lines 2-4 and Figure S2: Could the authors explain the difference they make between MDS (in the text) and CDS (in Figure S2)? My understanding is that a CDS is the entire gene coding sequence and may be made of multiple MDSs. If this is correct, the sentence should read "We compared the number of MDSs between single-copy orthologs for single-gene MAC chromosomes across the three species and found that the orthologs have similar CDS lengths".

Yes, we made the correction.

4. p. 12 lines 10-15: the discovery that paralogous MDSs can be found in scrambled genomic loci is interesting. If the two paralogs can be distinguished based on the number of substitutions, it would be informative to go back to individual reads and check whether each of the two copies can be incorporated in the unscrambled CDS (and at which frequency). Would the pointers be compatible with this?

The paralogous MDSs in the MIC are often not identical. The copy with the highest similarity is assigned as “preliminary match” by SDRAP (ref. 52), and others are assigned as “additional matches”. To validate SDRAP assignments, we did pairwise BLASTN alignments (“-task megablast”) of paralogous MIC MDSs and their corresponding MAC MDSs. We confirmed that in the three species, the preliminary match has the best or equally best pid (percentage of identity) in most cases. Therefore, the MDS assigned as preliminary match is more likely the paralog incorporated into the MAC chromosome.

We used genome assemblies of *Euplotes woodruffi*, which had the highest Nanopore coverage, to further investigate the frequency of MDS incorporation. We followed the reviewer’s suggestion and called SNP variants on both MAC and MIC genomes. For MAC SNP calling, we used Illumina reads as input for freebayes (ref a). For MIC SNP calling, we used Nanopore reads, instead of Illumina reads, to avoid non-specific short-read mapping on paralogous MDSs and to avoid the presence of any contaminating MAC reads. Variants were called and phased by PEPPER-Margin-DeepVariant (ref b), a new tool published in 2021 in Nature Methods, which has been reported to have similar accuracy to Illumina read variant calling, especially at high read coverage. We used the parameter “--pepper_min_coverage_threshold 20” to call confident variants when at least 20 reads cover the position. Only 92 MIC SNPs in the paralogous MDSs passed all filters of the program. Using this small set of MIC SNPs, we were unfortunately unable to distinguish which paralogous MIC MDS was incorporated into the MAC. Therefore, we cannot infer with what frequency one paralogous MDS is incorporated over another, until they become sufficiently diverged, which is compatible with the model.

Garrison E, Marth G. Haplotype-based variant detection from short-read sequencing. arXiv preprint arXiv:1207.3907. 2012 Jul 17.Shafin K, Pesout T, Chang PC, Nattestad M, Kolesnikov A, Goel S, Baid G, Kolmogorov M, Eizenga JM, Miga KH, Carnevali P. Haplotype-aware variant calling with PEPPER-Margin-DeepVariant enables high accuracy in nanopore long-reads. Nature methods. 2021 Nov;18(11):1322-32.

5. The hypothesis that odd-even scrambled loci have evolved from paralogous genes in E. woodruffi is supported by the existence of paralogous MDSs, length conservation of MDS/IES pairs and sequence similarity between corresponding MDS and IES in a pair. The correlations presented for Oxytricha and Tetmemena are much less convincing (Figure S5D and E). I recommend that the authors are even more cautious in their statement on p.13 ("For Oxytricha and Tememena, the MDS and IES lengths for such MDS/IES pairs also correlate positively, but more moderately")

Thank you, we rephrased the text.

6. p. 15 last paragraph: Why did the authors focus only on TBEs inserted in non-scrambled IESs to look for orthologous TBE insertions? Is there a reason to believe that no recent TBE insertion occurred at other genomic loci? Or was it only for practical reasons? It is also not clear to me whether the authors have considered full-length TBEs or the presence of at least one TBE ORF.

This analysis was limited for practical reasons, because we identify position conservation of TBEs by aligning protein sequences of MAC genes. We only consider TBEs inserted in non-scrambled IESs in exons. It would be difficult and less meaningful to align completely non-coding MIC-limited regions.

Partial TBEs are also included if they contain at least one TBE ORF (detected by BLAST).

Furthermore, TE insertion cannot explain the origin of scrambled IESs, and TEs rarely map to scrambled IESs (Figure S1A), but there is a clear evolutionary model for the origin of nonscrambled IESs from decay of TBEs (ref. 49). Initial purifying selection would act on the TE to maintain its ability to self-excise, whereas we advocate for a different model for the origin of scrambled IESs by decay of paralogous MDSs.

7. p. 16: the authors report that some introns of E. woodruffi map "near" Oxytricha/Tetmemena pointers. How near? Based on the information provided by the authors, I don't think this observation necessarily implies that IESs were converted to introns (or reciprocally) during evolution. If this were true, shouldn't at least one intron boundary coincide exactly with a pointer? The authors should clarify this (also in the discussion, on p. 20, top paragraph).

We used a 20bp window (~7 amino acids), as described in the Methods, and added that to the Results. Full detail is provided in the Methods section, “Ortholog comparison pipeline and Monte Carlo simulations”. 103 *E. woodruffi* introns are within 20bp from the midpoint of *Oxytricha/Tetmemena* pointers. Among these, 43 intron boundaries overlap an *Oxytricha* or *Tetmemena* pointer. We observed 306 cases of precisely matching boundaries between any two species, where the exon junction of one species maps inside the MDS/IES pointer of another species, although we would only expect the boundaries of introns and IESs to coincide so precisely if they were recent conversions. Hence we feel that a window analysis is informative.

8. p. 19 2nd paragraph: the suggested mechanism explaining the 5' bias of IESs in E. woodruffi genes is unclear. How could germline recombination take place between a MIC chromosome and a MAC reverse transcript or nanochromosome? This would imply that DNA could be imported in the MIC. Is there evidence that this might occur?

The ability of TEs to invade the MIC demonstrates that even foreign DNA can be incorporated into the MIC. Since MAC DNA is present at high copy number, it offers a potential source for a recombination template that could erase IESs, as could an errant reverse transcript of one of the long noncoding template RNAs. Any of these would be infrequent events that would matter on an evolutionary time scale even if developmentally rare.

9. According to Figure 1, no scrambled genes have been reported in Paramecium tetraurelia. Within the frame of the proposed model, this is somewhat unexpected because this ciliate went through several whole genome duplications during evolution and harbors many paralogous gene pairs. Is there a reason why no gene scrambling took place in Paramecium?

*Paramecium* uses only TA dinucleotide pointers for IES elimination, unlike the rich diversity of pointers in spirotrichous ciliates. This limitation in its machinery may explain why no scrambled loci have been observed in *Paramecium*, despite the abundance of paralogs. Our model suggests that local MIC paralogy is associated with the origin of scrambling. But most of the paralogy reported in *Paramecium* is at the level of whole chromosomes in the MAC (ref. 104) rather than local MIC paralogy.

Minor comments:p. 4 (4th bottom line): To my knowledge, ref #28 presents a draft (incomplete) MIC assembly of the Paramecium genome.

Thank you, we added reference 29 and adjusted the wording describing the quality of MIC genome draft assemblies.

p. 7 (last paragraph): "encoding" should be replaced by "carrying"

Thank you, we made the change.

p. 10 (2nd paragraph): insert a missing "o" into "nanochromosomes"

Thank you, corrected.

p. 10 (same paragraph): the weak 5' bias of IES distribution in Tetmemena should be shown (either as an additional panel in Figure 3 or in a Sup Figure).

Thank you, we added it as Figure S2C.

p. 24 2nd paragraph: "a" is missing in "Trinity, which is a software…"

Thank you, we made the correction.

Cross-Consultation CommentsI agree with most comments of reviewer 3.The authors have actually defined "TE" in the introduction (p. 6). Depending on the journal's rules for abbreviation use, it may not be necessary to define it again in the Results sectionReviewer #1 (Significance (Required)):Ciliates are unicellular models to study developmentally programmed genome rearrangements at the mechanistic, genome-wide and evolutionary levels. These aspects have so far mostly been addressed in three species: *P. tetraurelia* and *Tetrahymena thermophila* on the one hand, the spirotrichous ciliate O. trifallax on the other.One new piece of information that can be found in the present manuscript is the assembly and annotation of the germline genome of two novel species: Tetmemena sp, closely related to Oxytricha, and the more distant E. woodruffi. Feng et al. establish that, similar to other ciliates, Tetmemena and Euplotes eliminate TEs and other germline-specific sequences during programmed genome rearrangements. They also undergo extensive gene unscrambling, which results in IES removal and MDS reordering to assemble coding sequences.A TE origin was discussed previously for Paramecium (Arnaiz et al. PLoS Genet; Sellis et al. 2021 PLoS Biol) and Tetrahymena IESs (Hamilton et al. 2016 eLife). While this may also hold true in spirotrichous ciliatesThe present manuscript proposes a completely new evolutionary scenario for IESs from scrambled genes. Here, Feng et al. establish that scrambled genes of spirotrichous ciliates tend to be associated with local paralogy. They provide evidence supporting that IESs from scrambled genes may have evolved from paralogous MDSs.Although I am more an expert in the molecular mechanisms involved in genome rearrangements, I feel that the work reported here should draw the attention of a broader audience interested in genome dynamics and evolution, beyond the specific field of spirotrichous ciliate biology.Reviewer #3 (Evidence, reproducibility and clarity (Required)):Feng et al. provide a solid analysis of the evolution of genome rearrangement in spirotrich ciliates. The authors applied a variety of state-of-the-art sequencing and bioinformatic methods to investigate the intriguing and extremely complex patterns of genome architecture in this protist lineage. Methods (including statistical analyses) are adequate and explained in detail. Results and discussions reflect careful, clever analysis of the data and excellent linkage with the literature. Figures and tables complement the text in a compelling way. I have only minor suggestions:Summary: more gradually introduce Spirotrichea and the phylogenetic relationship among the three species analyzed. This would better position the reader to understand the evolutionary context you are working in. Also, it would be helpful to more clearly differentiate novel vs. existing data. A suggestion: "This study focuses on three spirotrich species: two in the family Oxytrichidae (Oxytricha trifallax and Tetmemena sp) and Euplotes woodruffi as an outgroup. To complement existing data, we sequenced, assembled and annotated the germiline and somatic genomes of E. woodruffi and the germline genome of Tetmemena sp."

Thank you, we clarified the summary (abstract).

Introduction, first paragraph: Replace "The species in this study…" for a more precise statement, such as "The three spirotrich species studied here…"

Thank you, we have made this statement more precise.

p. 4: This sentence is unclear: "These useful tools provide partial insight to guide selection of species for full genome sequencing, which allows construction of complete rearrangement maps of a MIC genome onto a MAC genome for a reference species."

Thank you, we have clarified this sentence.

p. 8: define TE on first mention.

Defined on page 6.

Table 1. Indicate which MIC and MAC data are from this study.

References are included for published data and a note has been added to indicate data from this study.

Reviewer #3 (Significance (Required)):The present work represents a significant advance in the field of evolutionary genomics. The focus of the paper is on ciliates, an ancient (2 billion-year old) and highly diverse eukaryotic phylum that presents many peculiarities, including sex, nuclear dimorphism, genome rearrangement, high numbers of paralogs and transposons, etc. While some data exist on a few model ciliates of disparate phylogenetic position, this work focuses on two species taxonomically placed in the same family, plus a more distant outgroup within the same class. This gives a novel dimension to this study, that goes beyond exploring genome architecture in a single clade. Instead, it allows to explore evolutionary trends in genome rearrangement among relatively closely related species. This paper should be of high interest not only for ciliate biologists (like me), but also in relation to comparative genomics of protists/eukaryotes and germ-soma biology. I highly recommend publication.

[Editors' note: further revisions were suggested prior to acceptance, as described below.]

A major concern is that the genome assemblies are poor. Much of this is probably because of the technical challenges associated with this system, but BUSCO scores of 76% are and N50 <30kb for the MICs are red flags. This does make one wonder how confident one can be that some of the conclusions, such as those pertaining to paralogs, are not at least in part due to assembly artifacts. It is striking that most genes undergoing unscrambling have no orthologs in others species, and that is thus unclear what these genes actually do. Please provide additional evidence that the conclusions are unaffected by the limitations of the assemblies. Please also provide more statistics regarding the assemblies (e.g., expected lengths distribution of nanochromosomes in MACs, number of chromosomes in MICs vs contigs, etc.), and annotations (gene families -- how similar are they between species, how many "species-specific" genes [which the genomics community considers to be largely, though not entirely artifactual] etc.).

In this study, we compared both somatic MAC and germline MIC genomes in the three ciliate species. We would like to explain why we think all the genome assemblies are sufficiently high quality for our analysis.

The MAC genome of *Euplotes woodruffi* was newly assembled in this study, following the similar assembly pipeline for *Oxytricha trifallax* (6) and *Tetmemena sp.* (7). The BUSCO score of 76% that we reported in our submitted manuscript was assessed using BUSCO v3 and lineage dataset eukaryota_odb9. Fortunately, BUSCO recently updated their database (88) to sample more species including ciliates (see below). Using the latest BUSCO v5.4.3 and lineage dataset alveolata_odb10, we now find that the MAC genome of *E. woodruffi* has a BUSCO score of 88.8%, the highest for any published *Euplotes* genome. Therefore, we are confident in the quality of our MAC genome assembly.

As the authors of BUSCO describe (87), the BUSCO score is just one metric of assessing genome completeness and is strongly influenced by (1) the gene prediction model (the nonmodel ciliate *Euplotes* uses a different genetic code, UGA=Cys, and has fewer curated genes compared to other well-established model ciliates; therefore it has a less mature gene prediction model) and (2) the genetic distance from the sampled species in the BUSCO dataset (e.g. the *C. elegans* genome only contains 90% of BUSCO genes, as shown in ref. 87). In our case, the missing detection of BUSCO genes is more likely the result of the evolutionary distance between ciliates and the species used in the previous BUSCO dataset.

We also assessed other ciliate genomes for comparison (see Figure 1, below). The MAC genomes of four reference hypotrich ciliates (*Tetmemena sp., Laurentiella sp., Paraurostyla sp.* and *Urostyla sp.,* ref. 7), which were previously assessed as "complete" using core eukaryotic genes (CEG), have current BUSCO scores ranging from 88.3% to 94.1% (Figure 1). *Urostyla sp.*, which is the most diverged from *Oxytricha trifallax* (which is included in the current BUSCO reference dataset), has the lowest BUSCO score among those, consistent with the greater evolutionary distance, and does not reflect a less complete genome. Note that some ciliate genomes have BUSCO scores near 100% because they were included in the BUSCO dataset. Since no *Euplotes* species were sampled in the BUSCO dataset, we conclude that 88.8% is within the range of a high-quality ciliate MAC genome assembly.

**Author response image 1. sa2fig1:** BUSCO assessments of MAC genomes in representative ciliates. The three species in the present manuscript are shown in bold. Species with a * are expected to have high BUSCO scores because they were sampled in the BUSCO reference dataset (alveolata_odb10). MAC genomes included here: Euplotes octocarinatus (8), Euplotes vannus (9), Laurentiella sp., Paraurostyla sp., Urostyla sp. (shown to be complete based on the presence of core eukaryotic genes, 7), Halteria grandinella (Zheng et al., Genbank RRYP01000000; https://journals.asm.org/doi/10.1128/mBio.01964-20), Paramecium tetraurelia (https://paramecium.i2bc.parissaclay.fr/download/Paramecium/tetraurelia/51/annotations/ptetraurelia_mac_51/), *Tetrahymena thermophila* (http://ciliate.org/index.php/home/downloads).

We also provided in the Methods section two other commonly used metrics to assess MAC genome completeness, as in Swart et al (6) and Chen et al. (7): 80.6% of *E. woodruffi* MAC contigs contain at least one telomere. Furthermore, the *E. woodruffi* MAC genome contains tRNAs for all 20 amino acids. Both assessments support the conclusion that the *E. woodruffi* MAC genome assembly is of high quality.

The short N50 of MAC genomes is expected for ciliate MAC genomes with gene-sized chromosomes. We added Figure 2 - figure supplement 2 to the manuscript to show the length distribution of MAC nanochromosomes (telomere-to-telomere assembled contigs). This is itself a slight underestimate of the expected MAC length distribution, because the longest MAC chromosomes have a lower probability of complete assembly. It was also expected that *Euplotes* species would have shorter MAC nanochromosomes, compared to *Oxytricha* and *Tetmemena*, based on prior gel electrophoresis (Swanton, Greslin, and Prescott. 1980. *Chromosoma* 77:203215), which reported that the distribution of MAC DNA molecules in *Euplotes aediculatus* appeared to migrate faster than other hypotrichs (see below, Figure 2)*.*

**Author response image 2. sa2fig2:** Agarose gel electrophoresis visualizing the size distribution of MAC DNA from Euplotes aediculatus, Stylonychia pustulata (also known as Tetmemena pustulata) and Oxytricha nova. Figure 8 in ref. 13 and Swanton, Greslin, and Prescott. 1980. Chromosoma 77:203-215.

MIC genomes, on the other hand, are more challenging to assemble because of the high content of repetitive elements that hinder assembly (Figure 2 - figure supplement 1). In this study, the major use of the MIC genome assembly is to infer MDS and IES annotations. We report that 90% of the MAC chromosomes are well covered (>90%) in the MIC genome (reported in Results). This level of completeness is comparable to that of the *Oxytricha trifallax* reference genome (1), and therefore is sufficient to permit comparison of rearrangement maps, including scrambled loci.

The gene annotation and ortholog analysis in this study is only performed on MAC genes. The MIC genome is not used for gene or paralog annotation because coding regions are contained in MDSs that are interrupted by IESs and/or scrambled. Therefore, the fragmentation of MIC contigs does not influence the quality of gene annotation or paralog/ortholog assignment. Paralogs were carefully analyzed to include only genes found on telomere-terminating contigs (G_4_T_4_G_4_ or C_4_A_4_C_4_), to be confident that they represent MAC chromosomes. Furthermore, the MAC genome assemblies were clustered to a level of sequence similarity of 95% in order to collapse and therefore exclude allelic differences. For the purpose of studying paralogous MDSs in the MIC genome, we only included paralogous MDSs that map to the same MIC contig. This excludes the possibility of conflating MDSs that map to different contigs. Therefore, the higher levels of paralogy that we report for scrambled genes (both orthogroup size, which entirely derives from the MAC assembly, and numbers of paralogous MDSs identified on MIC contigs) are not likely due to assembly artifacts but meet a stringent quality standard.

Details of gene families identified by OrthoFinder for scrambled and nonscrambled genes were provided in Supplementary File 4. We also provide a new Supplementary File 5 to summarize gene families detected by OrthoFinder for each pair of species. Gene annotations for the three species have been uploaded to https://knot.math.usf.edu//mds_ies_db/2022/downloads.html. Supplementary File 4 and 5 also provide an estimate of "species-specific" genes. We do not observe more species-specific genes among scrambled vs. nonscrambled loci (see the response to point 2 below).

We do not presently report the number of MIC chromosomes in any species, but this will be possible for *Oxytricha trifallax* from a new study based on Hi-C from another first author. Chen et al. (1) reported an estimate based on collapsing MIC-telomere-containing PacBio reads, but we find that Hi-C will provide a more accurate estimate.

Please spell out more clearly what you consider the evolutionary forces that have increased scrambling. My reading is that you consider many of the scrambled genes not to be functionally very important, which would be in line with there being so few orthologs in other species. To this end, please provide some basic information as to how many of the scrambled genes have expression support, what the distribution of expression quantiles is etc. -- this could help to shore up these inferences.

We do not suggest that scrambled genes are less important than nonscrambled ones. Supplementary File 4 shows that a similar portion of scrambled vs. nonscrambled genes have orthologs in other ciliates. The lack of orthologs in other ciliates may be due to less investigation of closely related species. This could explain why *Oxytricha* genes, whether scrambled or nonscrambled, have more orthologs detected in other ciliates compared to *Euplotes*.

Our model suggests that local duplication provides a buffer against mutations, allowing paralogous MDSs to repair the MAC locus during assembly of the scrambled genes. The increased levels of scrambling in the *Oxytricha* lineage do not need to invoke a fitness advantage, however. It is more likely that a neutral ratchet drives the increase in scrambling, because of the difficulty of "erasing" it from the germline MIC genome once a scrambled architecture has been established, relative to the trend towards shortening MDSs as more mutations accumulate in either paralog, which leads to increased levels of fragmentation (68, 69). We add this to the discussion.

We followed the editor’s suggestion to analyze gene expression data for all species. We collected poly-A enriched mRNA and sequenced three replicates from asexually growing vegetative cells. We find that scrambled and nonscrambled genes have nearly identical levels of expression support (at least one read in all 3 replicates) in both *Oxytricha* (Supplementary File 8) and *Tetmemena*. *E. woodruffi* has more expression support for nonscrambled vs. scrambled genes, on the other hand (Supplementary File 8), which could be due to the more recent acquisition of scrambled loci in *E. woodruffi*, based on the stronger correlations observed between MDS and IES length and sequence similarity for MDS/IES pairs flanked by the same pointers (Figure 4 - figure supplement 2). Thus, it is possible that their nonscrambled paralogs may still serve the major function.

We also compared the expression levels of scrambled vs. nonscrambled genes. Only genes with a low Coefficient of Variation (CV<1) of TPM (transcripts per million) are included in the comparison (new Figure 4 - figure supplement 3). The distribution of expression levels is similar for scrambled vs. nonscrambled genes (Figure 4 - figure supplement 3). In a Mann-Whitney U test, the average expression level among three replicates is significantly higher in nonscrambled genes for *Oxytricha* and *E. woodruffi*, but not significant for *Tetmemena*.

Finally, the manuscript is overly long, and despite -- or perhaps because -- of this, it is not presented in the most accessible manner. For example, the abstract still struggles to explain the importance or relevance of the work. e.g. the last sentence is "Scrambled loci are more often associated with local duplications, supporting a simple model for the origin of scrambled genes via DNA duplication and decay". This would need some motivation, i.e., why should we care about scrambled genes per se? Similarly, why does the Introduction have an almost 1-page summary of the results?

We provide additional motivation for the study of scrambled genomes in the beginning of the abstract and the introduction. *Oxytricha*'s scrambled genome and others in its lineage represent some of the most complex genome architectures known in *any organism*, with hundreds of thousands of precise, programmed genome editing events required to assemble coding regions. Recent exciting reports have described scrambled genomes in metazoa, including cephalopods, but those evolutionary events entail only shuffling of gene order, with no accompanying genome editing, so they are less complex, and other examples of programmed genome rearrangement in both protists and metazoa are usually simple DNA deletions.

We shortened other portions of the manuscript, eliminating or combining paragraphs where feasible, and added more motivation throughout, particularly emphasizing the compelling ways in which the *Oxytricha* lineage presents the most complex natural genome editing pathway known to date. Hence it is important to ask how such complex genomes have arisen, and comparative genomics provides fundamental insight into this question.